# Lipid Mediators Regulate Pulmonary Fibrosis: Potential Mechanisms and Signaling Pathways

**DOI:** 10.3390/ijms21124257

**Published:** 2020-06-15

**Authors:** Vidyani Suryadevara, Ramaswamy Ramchandran, David W. Kamp, Viswanathan Natarajan

**Affiliations:** 1Department of Pathology & Laboratory Medicine, Indiana University School of Medicine, Indianapolis, IN 46202, USA; visurya@iu.edu; 2Departments of Pharmacology & Regenerative Medicine, University of Illinois, Chicago, IL 60612, USA; ramchan@uic.edu; 3Department of Medicine, Division of Pulmonary & Critical Care Medicine, Jesse Brown VA Medical Center, Chicago, IL 60612, USA; d-kamp@northwestern.edu; 4Department of Medicine, Northwestern University Feinberg School of Medicine, Chicago, IL 60611, USA; 5Department of Medicine, University of Illinois, Chicago, IL 60612, USA

**Keywords:** pulmonary fibrosis, lipid mediators, sphingolipids, sphingosine-1-phosphate, sphingosine kinase 1, prostaglandins, lysophosphatidic acid, autotaxin, G-protein coupled receptors, lysocardiolipin acyltransferase, phospholipase D, oxidized phospholipids

## Abstract

Idiopathic pulmonary fibrosis (IPF) is a progressive lung disease of unknown etiology characterized by distorted distal lung architecture, inflammation, and fibrosis. The molecular mechanisms involved in the pathophysiology of IPF are incompletely defined. Several lung cell types including alveolar epithelial cells, fibroblasts, monocyte-derived macrophages, and endothelial cells have been implicated in the development and progression of fibrosis. Regardless of the cell types involved, changes in gene expression, disrupted glycolysis, and mitochondrial oxidation, dysregulated protein folding, and altered phospholipid and sphingolipid metabolism result in activation of myofibroblast, deposition of extracellular matrix proteins, remodeling of lung architecture and fibrosis. Lipid mediators derived from phospholipids, sphingolipids, and polyunsaturated fatty acids play an important role in the pathogenesis of pulmonary fibrosis and have been described to exhibit pro- and anti-fibrotic effects in IPF and in preclinical animal models of lung fibrosis. This review describes the current understanding of the role and signaling pathways of prostanoids, lysophospholipids, and sphingolipids and their metabolizing enzymes in the development of lung fibrosis. Further, several of the lipid mediators and enzymes involved in their metabolism are therapeutic targets for drug development to treat IPF.

## 1. Introduction

Lipids are the principal constituents of cell membranes and play an essential role in several physiological and pathophysiological processes by mediating intracellular and extracellular cues. Phospholipids and sphingolipids, which are the structural components of the membranes, regulate cell shape, ion transport, intra- and inter-cellular communication and signaling. Many of the lipid-derived mediators are short-lived second messengers, and regulate cellular functions as migration, proliferation, apoptosis, redox balance, and cytoskeletal organization. Alteration or aberration in the generation of lipids mediators has been shown to regulate the physiology and pathophysiology of several disorders including, but not limited to, cancer, brain injury, cardiovascular diseases, kidney diseases, and pulmonary complications [1,2,3,4,5,6]. This review will specifically highlight the involvement of lipid mediators in pulmonary fibrosis (PF) and provide some mechanistic insights into the regulation of idiopathic pulmonary fibrosis (IPF) pathology by various lipid metabolites in animal models that mimic IPF.

IPF is a progressive fibrotic disease of the lung of unknown etiology that occurs in older adults, diagnosed as usual interstitial pneumonia with a clinicopathologic criteria [7], wherein the lung tissue becomes thickened from scarring [8,9,10]. The compromised architecture leads to disturbed gas exchange, decreased lung compliance, and respiratory failure and death [11]. Recurrent injury to the lung epithelium triggers pro-inflammatory and pro-fibrotic signaling involving the alveolar epithelial cells (AECs), alveolar macrophages (AM), fibroblasts, and endothelial cells contributing to the fibrotic foci and progression of IPF [12,13,14,15]. This pathogenesis is characterized by epithelial cell apoptosis, epithelial-to-mesenchymal cell-transition (EMT), endothelial-mesenchymal transition (EndMT), activation of fibroblasts which differentiates into contractile myofibroblasts leading to deposition of the extracellular matrix and scar tissue formation [12].

TGF-β is a critical cytokine that drives development of fibrosis. In mammals, three major isoforms of TGF-β have been identified, namely TGF-β1, -2, and -3 [16], and TGF-β1 is the predominant isoform expressed in lungs of IPF patients and preclinical models [17]. Activation of TGF-β1 binding to TGF-βRII via the SMAD2/3/4-dependent pathway leads to the fibrogenic program with extracellular matrix synthesis. TGFβ1-mediation of PF is recognized additionally via SMAD-independent non-canonical pathways [18]. Some of these known regulators include JNK kinase, MAPKinase, PI3K/Akt, and Rho kinase pathways, and their inhibitors are being targeted for clinical interventions in IPF [19]. Inhibition of bleomycin-induced PF and extracellular matrix (ECM) deposition was demonstrated by imatinib, a tyrosine kinase inhibitor, suggesting a crucial role for cAbl kinase [20]. The inhibition of the lectin, galecin-3, presumably derived from alveolar macrophages has also been shown to diminish bleomycin and TGFβ-induced fibrosis in a SMAD-independent manner [21]. Currently, IPF has only two drugs, Nintedanib and Pirfenidone, approved by the Food and Drug Administration for treatment [12]. Unfortunately, these drugs do not cure the disease, but only aid in slowing the progression of the disease. Thus, there is a crucial requirement for identifying new targets and signaling pathways that underlie the mechanisms behind IPF [22]. This review is specifically focused on the role of lipid-derived mediators and their signaling pathways modulating pulmonary fibrosis, in humans, and preclinical models. It is beyond the scope of this review to touch on all aspects of the disease since several recent reviews do justice in this context and will also sway the subject away from lipid signaling [9,10,12].

## 2. Plasma Lipid Profile in IPF Patients

Aberrations in phospholipids and sphingolipids metabolism have been identified as potential contributors to the pathophysiology of IPF. A recent lipidomics study revealed that several lipids were found to be altered in plasma of IPF patients. Several glycerophospholipids that were screened in this study were found to be lowered in IPF patients. 30 out of the 159 glycerolipids were distinct between control and IPF patients [23]. These altered lipid profiles in plasma can be exploited as potential biomarkers for IPF.

### 2.1. Fatty Acids and Fatty Acid Elongation in Pulmonary Fibrosis

Biosynthesis of palmitic acid (C16:0) and other long-chain saturated and unsaturated fatty acids is central to the generation of triglycerides, phospholipids, and sphingolipids with different fatty acid molecular species that dictate their function and metabolic fate. Cellular levels of free fatty acids are very low in tissues; however, elevated levels of palmitic acid were detected in the lungs of patients with IPF compared with control subjects [24]. Palmitic acid-rich high fat diet-induced epithelial cell death and a prolonged pro-apoptotic endoplasmic reticulum (ER) stress response after bleomycin-induced lung fibrosis. Palmitic acid accumulation in IPF lungs could be due to a defect in long-chain fatty acid family member 6 (Elovl6) enzyme that converts palmitoyl CoA to stearoyl CoA or a defect in stearoyl CoA desaturase that converts palmitic acid to palmitoleic acid (C16:1 n-7) or stearic acid to oleic acid (C18:1 n-9). The expression of *Elovl6* was decreased in lungs of patients with IPF and lungs of bleomycin-treated mice [25]. *Elovl6* depletion in LA-4 epithelial cells increased the cellular levels of palmitoleic acid (C16:1 n-7) and decreased stearic acid (C18:0) whereas the levels of linoleic acid (C18:2 n-6) and arachidonic acid (C20:4 n-6) were unchanged. Further, depletion of *Elovl6* with siRNA or exogenous addition of palmitic acid to LA-4 epithelial cells increased reactive oxygen species (ROS) and apoptosis, which was inhibited by oleic or linoleic acid, while ROS generation was elevated in *Elovl6*^−/−^ mice [25]. Similarly, deficiency of stearoyl CoA desaturase-1, which catalyzes conversion of saturated to monounsaturated fatty acid, induced ER stress, and experimental PF in mice [26]. These findings suggest that lipotoxicity due to accumulation of saturated fatty acids may have a detrimental role in the development of lung fibrosis in IPF and in animal models of lung fibrosis by inducing ER stress and apoptosis in AECs.

### 2.2. Nitrated Fatty Acids in Pulmonary Fibrosis

Nitrated fatty acids (NFAs) are produced by non-enzymatic reactions between nitric oxide (NO), unsaturated fatty acids such as oleic acid, and linoleic acid to generate 10-nitro-oleic acid (OA-NO2), and 12-nitrolinoleic acid (LNO2), respectively [27,28,29,30]. OA-N2 and LNO2 are the most abundant NFAs in human plasma [27], and both are physiological activators of the nuclear hormone receptor peroxisome-activated receptor γ (PPARγ), which exhibits tissue-protective and wound healing properties [31]. PPARγ agonists have been shown to exhibit antifibrotic activity in vitro [32], and in bleomycin-induced PF [33]. NFAs upregulated PPARγ and blocked TGF-β-induced fibroblast differentiation in vitro and administration of OA-NO2, in mice, post-bleomycin challenge ameliorated and reversed bleomycin-induced PF, suggesting therapeutic potential of NFAs in resolving PF [34].

### 2.3. Prostaglandins and Leukotrienes in Pulmonary Fibrosis

Prostaglandins and leukotrienes are lipid autacoids derived by the action of cyclooxygenases (COXs) 1 and 2 and lipoxygenases, respectively, that oxygenate and cyclize arachidonic acid released from membrane phospholipids such as phosphatidylcholine by the action of phospholipase (PL) A_2_. The oxygenated arachidonic acid intermediate thus generated leads to production of prostaglandin (PG) E2 (PGE2), prostacyclin (PGI2), prostaglandin D2 (PGD2), prostaglandin F2α (PGF2α), thromboxane A2 and other eicosanoids catalyzed by a specific PG synthase [35]. Cytosolic PLA_2_ has been shown to play a key role in generating proinflammatory eicosanoids, and depletion of cytosolic PLA_2_ attenuated bleomycin-induced PF and inflammation by reducing production of leukotrienes and thromboxanes [36]. Since cytosolic PLA_2_ is the rate-limiting enzyme in eicosanoid biosynthesis, there has been an ongoing push to develop specific inhibitors for cytosolic PLA_2_ by researchers and Pharma companies. One such potent and selective inhibitor of cytosolic PLA_2_, AK106-001616, reduced the disease score of bleomycin-induced lung fibrosis in rats [37], but it is unclear if this inhibitor has been clinically tested for its efficacy against IPF.

### 2.4. PGD2 in Pulmonary Fibrosis

Prostaglandin D2 (PGD2) is synthesized by hematopoietic PGD synthase (H-PGDS) in hematopoietic linage cells including mast cells and Th2 lymphocytes. Depletion of H-PGDS was found to accelerate bleomycin-induced PF and increase vascular permeability [38]. This was found to be predominantly mediated by inflammation, as seen by increased expression of H-PGDS in the neutrophils and monocytes/macrophages of an inflamed lung [38]. Supporting this, it was shown that retroviral injection of H-PGDS expressing fibroblasts in the lung attenuated bleomycin-induced lung injury and fibrosis [39]. Though the direct role of PGD2 was not shown in PF, the impact of PGDS and PGD receptor has been identified. Genetic depletion of chemoattractant receptor homologous with T-helper cell type 2 (CRTH2), a receptor for PGD2_,_ in mice aggravated bleomycin-induced PF, as seen by prolonged inflammation and delayed resolution of fibrosis [40]. γδT cells expressing CRTH2 were found to be important in imparting protection against bleomycin-induced PF, when compared to splenocytes and other hematopoietic cells [40]. PGD2 also regulates fibroblast activity by attenuating TGFβ-induced collagen secretion via the DP receptor and the c-AMP pathway [41]. PGD2 was also found to mediate IPF by triggering of MUC5B gene expression in airway epithelial cells through the activation of ERK MAPK/RSK1/CREB pathway by binding to D-prostanoid receptor (DP1) [42]. MUC5B plays a key role in the development of honeycomb cysts seen in the lungs of IPF patients. Thus, activation of H-PGDS or depletion/inhibition of CRTH2 in specific cell types may be a therapeutic approach in ameliorating experimental PF in animal models. It is unclear if blocking PGDS or CRTH2 in IPF patients has any beneficial impact on human lung fibrosis.

### 2.5. PGE2 and PGE2 Signaling in the Pathogenesis of Pulmonary Fibrosis

Prostaglandin E2 (PGE2), a COX-2-derived eicosanoid, plays an important role in regulating homeostatic signaling between AECs and lung fibroblasts; however, evidence for its profibrotic role in animal models is controversial. PGE2 levels in BAL fluids from IPF patients and bleomycin-treated mouse lungs are lower compared to normal subjects and mice not challenged with bleomycin. While most of the in vitro studies show that COX-2 and PGE2 are anti-fibrotic, the in vivo data have been inconsistent in animal models of lung fibrosis. In a vanadium pentoxide-induced model of PF, mice lacking COX-2 exhibited higher inflammatory responses and lung fibrosis compared to controls or mice lacking COX-1 [43]. However, in another study, COX-2 deficient mice had exacerbated lung dysfunction but not fibrosis to bleomycin challenge [44], while a different study showed that *COX-2^-/-^* mice developed both losses of pulmonary function and severe lung fibrosis to bleomycin challenge [45]. Reasons for these discrepancies are unclear but could be due to differences in the sex and genetic background of the mice, as well as the dose and route of bleomycin administration. Further, there have been controversies over the efficacy of PGE2 vs. PGI2 in conferring protection against bleomycin-induced PF in animal models. Treatment with PGE2 in the lung was found to improve the bleomycin-induced decline in the lung function and the inflammatory responses in the lung caused by bleomycin. However, there was no change in the extent of bleomycin-induced fibrosis in the mice treated with PGE2 [45]. A robust protection against bleomycin-induced lung injury and fibrosis was achieved by a nanostructured lipid carrier-based delivery of PGE2 specifically to the lungs along with siRNA targeted against CCL2, HIF1α, and MMP2. This was found to reduce the tissue damage, inflammation, fibrotic markers in the lung, in addition to improving the mortality [46]. Liposomal instillation of PGE2 through inhalation, but not intravenous injection protected mice against bleomycin-induced inflammation, reduction in body weight, extent of fibrosis, and mortality rates [47]. Losartan, a selective AT1 receptor antagonist, which increases PGE2 levels in the lung was found to improve bleomycin-induced inflammation and also reduce the hydroxyproline content in the lungs [48]. However, using several in vivo genetic models it was demonstrated that prostacyclin (PGI2), but not PGE2, protected against the development of fibrosis and decline in lung function in response to bleomycin treatment [49]. Derivatives of PGE2 have been tested for protection against lung fibrosis. Administration of 16, 16-dimethy PGE2 (DM PGE2) protected mice against bleomycin-induced lung inflammation and PF [50]; however, the effect of DM PGE2 on pulmonary function was not examined. In addition to the direct role of PGE2 in modulating fibrosis, the PGE2 transporter, PGT/SLCO2A1, may also play a pathophysiological role in the development and progression of fibrosis. Extracellular PGE2 is taken up by cells via the high-affinity transporter *SLCO2A1*, which is expressed in the vascular endothelium, human and mouse airway bronchial and alveolar Type I and Type II epithelial cells [51]. Mice deficient in *Slco2a1* exhibited more severe fibrosis to bleomycin administration as characterized by exacerbated collagen deposition as compared to WT mice [51]. The mechanism for the protection conferred by SLCO2A1 is unclear and it is not known why blocking PGE2 transport from outside to inside the cell exacerbated the fibrosis in bleomycin-challenged mice. Several unanswered questions such as expression of PGE2 receptors (EP2 and EP4), levels of PGE2 and TGF-β, and PGE2 signaling in *Sclco2a1* deficient cells in mouse lung need to be addressed to define the potential mechanism(s) of regulation by the transporter.

The current understanding of potential mechanisms involved in PGE2-mediated attenuation of lung fibrosis and fibroblast to myofibroblast differentiation could involve but not limited to: (1) PGE2 deficiency in fibrotic lungs; (2) modulation of PGE2 signaling via EP1-4 receptors regulating apoptosis and proliferation; (3) epigenetic regulation of EP receptors in fibrotic lungs; (4) Interaction between plasminogen activation and PGE2 generation; and (5) Modulation of cross-talk between PGE2/EP2/EP4 and TGF-β/TGF-βR1/TGF-βR2 signal transduction (Figure 1).

#### 2.5.1. PGE2 Deficiency in Fibrotic Lungs

Reduced PGE2 levels have been reported in bronchoalveolar lavage fluid and conditioned culture media of alveolar macrophages (AMs) from IPF patients [52,53], which are consistent with reduced COX-2 expression in IPF lungs [54,55,56]. Reduced PGE2 synthesis has been observed in fibroblasts isolated from bleomycin-challenged rat lungs [57,58,59] due to diminished expression of COX-2. This reduction in COX-2 expression in fibroblasts from IPF lungs was attributed to the inability of transcriptional factors such as NF-κB/p65, CEBPb, and CREB-1 to bind to COX-2 promoter due to a defective H3 and H4 histone acetylation resulting from increased recruitment of HDACs and decreased HATs [54]. Injury to the AECs during lung fibrogenesis resulted in an elevated release of chemokine CCL2 [59], which reduced PGE2 production and stimulated fibrogenesis [60].

#### 2.5.2. Modulation of PGE2 Signaling via EP1-4 Receptors Regulating Apoptosis and Proliferation

The cellular membrane receptors for PGE2 are termed EP receptors that consist of four different isoforms namely EP1, EP2, EP3, and EP4 generated from a primitive PGE receptor by gene duplication [61]. These four EP receptor isoforms show varying degrees of binding to PGE2 with EP3 and EP4 exhibiting high-affinity binding to the ligand [62]. In the normal lung, PGE2 secreted by AECs inhibits apoptosis of the cells in an autocrine fashion via EP2. However, PGE2 secreted by AECs limit fibroblast proliferation via EP2/EP4, activate PTEN, increase cAMP levels via adenylate cyclase that result in diminished fibroblast proliferation. Further, the ability of PGE2 to promote normal fibroblast apoptosis requires signaling through EP2/EP4 and reduction in protein kinase B (Akt) [63]. Akt is also negatively regulated by PGE2 by increased cAMP levels generated by PGE2 via EP2/EP4 and PTEN. PGE2 also limits myofibroblast differentiation in normal lungs by amplifying the inhibitory cAMP and PTEN [64]. cAMP generated by adenylate cyclase is rapidly degraded in cells by phosphodiesterase (PDE) 4, and blocking PDE4 that results in accumulation of cAMP induced by PGE2 also limits TGF-β-induced myofibroblast differentiation [65,66]. In contrast to the anti-proliferative effect of PGE2 in lung fibroblasts via EP2/EP4, PGE2 can promote proliferation of NIH 3T3 fibroblasts by promoting calcium mobilization [67]. Similarly, PGE2 stimulated neonatal rat ventricular fibroblast proliferation that was mimicked by sulprostone, an antagonist of EP1/EP3 [68]. Thus, PGE2 is a double-edged sword that can be antifibrotic or proliferative depending upon the type of EP receptor signal transduction and nature of the fibroblast investigated.

#### 2.5.3. Epigenetic Regulation of EP Receptors in Fibrotic Lungs

In addition to diminished PGE2 production and signaling in fibrotic lungs, fibroblasts from IPF lungs [69], and mice with experimental fibrosis [70] showed resistance to antifibrotic activities of PGE2 due to decreased expression of EP2 (PTGER2). The decreased expression of EP2 could be due to increased degradation, decreased synthesis, or aberrations in DNA methylation. Studies carried out with IPF patients and two animal models of PF identified hypermethylation of the human *PTGER2* and mouse *Ptger2* promoters, respectively, containing abundant CpG dinucleotides susceptible to methylation [71]. Further, inhibition of DNA methylation with 5-aza-2′-deoxycytidine restored EP2 mRNA and protein expression and responsiveness to PGE2 in IPF lung fibroblasts. The increased *PTGER2* promoter methylation was mediated by increased Akt signaling and *PTEN* suppression in PGE2 resistant IPF lung fibroblasts [71]. Thus, epigenetic hypermethylation of EP2 provides a novel mechanism of conferring resistance to PGE2 signaling in IPF lung fibroblasts, and in experimental animal models of fibrosis.

#### 2.5.4. Inter-relationship between Plasminogen Activation and PGE2 Production in Pulmonary Fibrosis

Plasminogen activation to plasmin protects lungs from fibrosis and patients with PF exhibit fibrin accumulation in the lungs due to increased expression of plasminogen activation inhibitor-1 (PAI-1) [72,73]. One potential mechanism underlying the antifibrotic effect of plasmin is through PGE2. Plasminogen activation upregulates PGE2 in AECs, fibroblasts, and fibrocytes from control and bleomycin-treated mice, and PGE2 production was exaggerated in lung fibroblasts, fibrocytes and AECs from *Pai1^−/−^*mice compared to the cells from the control group. Further, it has been shown that plasmin stimulated PGE2 production in AECs, and fibroblasts involved the enzymatic release of HGF by plasmin and subsequent HGF-mediated upregulation of COX-2 [74]. Interestingly, PGE2 also modulates expression of the plasminogen activation system such as PAI-1 in non-lung cells [75,76]. These findings suggest an important inter-relationship between plasminogen activation to PGE2 and PGE2 stimulation of PAI-1 regulates lung fibrosis process.

#### 2.5.5. Crosstalk between PGE2 and TGF-β Signaling in Fibroblast Differentiation

PGE2 is an antifibrotic lipid mediator and TGF-β is a multifunctional cytokine that drives fibrosis in IPF and experimental models of lung fibrosis. Evidence strongly suggests potential interaction and crosstalk between PGE2/EP2/EP4 and TGF-β/TGF-βR1/TGF-βR2 signaling pathways in regulating anti- and profibrotic cascades in lung fibroblasts. It has been reported that TGF-β1 induces PGE2, but not procollagen synthesis in human fetal lung fibroblasts that was pertussis-toxin sensitive [77]; however, other studies have demonstrated that TGF-β1 downregulates COX-2 expression leading to decreased PGE2 in human lung cancer A549 epithelial-like cells, which is involved in fibrotic response to TGF-β1 [78]. Several studies have shown that PGE2 antagonizes TGF-β signaling and responses in lung fibroblasts. PGE2 inhibited TGF-β1-induced fibroblast-to-myofibroblast differentiation that was SMAD-independent but involved cell shape and adhesion-dependent binding. PGE2 had no effect on TGF-β1-mediated SMAD phosphorylation or its translocation to the nucleus but diminished phosphorylation of paxillin, STAT-3, and FAK and limited activation of PKB/Akt pathway [79]. Interestingly, PGE2 not only prevented but also reversed the TGF-β1-induced myofibroblast differentiation as characterized by the ability of PGE2 to reverse expression of 368 genes upregulated and 345 genes down-regulated by TGF-β1 [80]. Moreover, PGE2 inhibited TGF-β-induced mesenchymal-epithelial transition [81] suggesting modulation of the injury-repair process. Many of the opposing effects of PGE2 to TGF-β response seem to involve changes in intracellular calcium. In fibroblasts isolated from normal lungs, PGE2 inhibited TGF-β-promoted [Ca^2+^] oscillations and prevented the activation of Akt and Ca^2+^/calmodulin-dependent protein kinase-II (CaMK-II) but did not prevent activation of Smad-2 or ERK. PGE2 also eliminated TGF-β-stimulated expression of collagen A1, and α-smooth muscle actin (α-SMA) [82]. However, as fibroblasts isolated from IPF lungs show resistance to TGF-β and PGE2 signaling, it is unclear if the expression of EP2/EP4 and TGF-β R1/TGF-βR2 are both modulated in these differentiated myofibroblasts. Further studies are necessary to define the precise interaction between these two signaling pathways in the regulation of fibrogenesis.

### 2.6. PGF2α and PGF2α Receptor in Pulmonary Fibrosis

Prostaglandin F2α (PGF2α) has been shown to be a pro-fibrotic eicosanoid that stimulated fibroblasts proliferation and collagen production in a TGF-β independent manner. In vivo studies revealed that signaling via its cognate receptor *Ptgfr* is involved in the development of lung fibrosis, with attenuation of bleomycin-induced fibrosis but not inflammation in *Ptgfr^−/−^ m*ice and was independent of the TGF-β pathway [83]. Pharmacological inhibition of TGF-βR1 kinase in *Ptgfr^−/−^* mice further inhibited lung fibrosis suggesting that PGF2α/PF pathway was signaling in a TGF-β independent manner. Further, PGF2α and TGF-β signaling were synergistic in mediating increases in proliferation and collagen synthesis by murine and human fibroblast cell lines. PGF2α-induced collagen gene transcription was Rho kinase-dependent; however, Rho-kinase was not involved in TGF-β-mediated collagen transcription. Additionally, bronchoalveolar lavage (BAL) fluid from IPF patients had high levels of PGF2α when compared to BAL fluid from patients with sarcoidosis [83].

The clinical relevance of PGF2α in PF was investigated by measuring levels of 15-keto-dihydro PGF2α, a major and stable metabolite of PGF2α in plasma of IPF patients. Plasma concentrations of 15-keto-dihydro PGF2α were significantly higher in IPF patients than controls, which correlated with forced expiratory volume in 1 s, forced vital capacity, diffusing capacity for carbon, the composite physiologic index, 6 min walk distance, and end-exercise oxygen saturation [84]. Thus, an association of plasma PGF2α metabolite, 15-keto-dihydro-PGF2α with disease severity of IPF and prognosis, supports a potential pathogenic role for PGF2α in human IPF.

### 2.7. Leukotrienes and Its Role in Pulmonary Fibrosis

Leukotrienes are immuno-regulatory lipid mediators primarily derived from arachidonic acid by 5-lipoxygenase (5-LO) [85]. 5-LO converts arachidonic acid to 5-hydroperoxy eicosatetraenoic acid (5-HPETE), and then to leukotriene LTA_4_. LTA_4_ is converted to the dihydroxy fatty acid leukotriene LTB_4_ or conjugated to glutathione to generate LTC_4_. LTC_4_ is converted to LTD_4_ and LTE_4_ by sequential peptidolytic cleavage [85]. The leukotrienes LTC_4_, LTD_4_, and LTE_4_ are collectively known as cysteinyl leukotrienes (cysLT) and play an important role in airway diseases. Leukotrienes signal through G-protein coupled receptors (GPCRs), namely BLT1 and BLT2 for LTB_4_, and CysLT1, CysLT2, and CysLTE, also known as gpr99, for cysLTs [86]. Leukotrienes, initially identified to be generated by leukocytes, are also produced by mast cells, eosinophils, macrophages, and inflammatory cells [85]. LTB_4_ and LTC_4_ levels are elevated in BAL and lung tissue lysates from IPF patients, suggesting constitutive activation of 5-LO in IPF [85,87]. AMs were the major source of increased LTB_4_ and LTC_4_ levels in IPF lungs [88]. In animal models such as bleomycin- and silica-induced PF, the tissue and BAL fluid content of cysLTs and LTB_4_ was significantly elevated [89]. Furthermore, targeted disruption of 5-LO also attenuated bleomycin-induced injury as determined by reduction in the lung content of hydroxyproline [90]. Additionally, in the silica-induced model of PF, the expression of the LTB_4_ receptor was increased while the expression of CysLT type 2 receptor was downregulated in lung tissue. Furthermore, strong immunohistochemical staining for the CysLT type 1 receptor, but not CysLT type 2receptor, was observed in pathological lesions [89]. These findings suggest that an increase in LT production in the lung and modulation of the cysLT receptors may contribute to the progression of PF. The importance of cysLT receptors in the pathogenesis of PF was further revealed by downregulation of LTC_4_ synthase, a key enzyme in cysLT biosynthesis. Genetic deletion of LTC_4_ synthase protected mice from bleomycin-induced alveolar septal thickening by macrophages and fibroblasts and collagen deposition [91]. In contrast, knockdown of the cysLT1 receptor significantly increased both, the concentration of cysLTs in BAL and the magnitude of septal thickening. These findings provide strong evidence for cysLT1 in regulating bleomycin-mediated lung fibrosis and a shift in the homeostatic balance from cysLT1 to cysLT2 to drive the fibrogenesis [92]. However, the clinical relevance of these pathways in patients with IPF and other forms of PF are unknown.

#### Cellular Senescence and Leukotriene Metabolism in Pulmonary Fibrosis

IPF is a lung disorder of the elderly population, and there is compelling evidence for aging-associated factors such as cellular senescence, telomerase attrition, and dysregulated metabolism in the pathogenesis of lung fibrosis. Accumulation of senescent cells in the fibrotic tissue has been linked to the severity of IPF, and earlier studies have shown that modulation of 5-LO and COX-2 in senescent fibroblasts [93,94]. In vitro, induction of senescence in human lung fibroblasts (IMR-90) using irradiation increased expression of phospho-5-LO/total 5-LO ratio and secretion of cysLTs [95]. Further, the cysLT-rich conditional medium of senescent IMR-90 fibroblasts induced pro-fibrotic signaling in naïve fibroblasts, which was abrogated by inhibition of 5-LO. In a preclinical model of lung fibrosis, pre-treatment with ABT-263 that selectively eliminates senescent cells attenuated bleomycin-induced PF [96]. Interestingly, senescent fibroblast from IPF lungs, but not normal lungs, secreted cysLTs and not the antifibrotic PGs_1_ suggesting a role for senescent lung fibroblasts in contributing to a pool of cysLTs regulating the development of fibrosis [95].

## 3. Sphingolipids, Sphingolipid Metabolizing Enzymes, and S1p Receptors in Pulmonary Fibrosis

Sphingolipids are present across all eukaryotic cells and serve as structural and signaling lipids that regulate a variety of cellular functions under normal and pathological conditions. Advances in comprehensive lipid profiling techniques using LC-MS/MS and LC-ESI-MS/MS have identified close to 400 sphingolipidome in human plasma and tissues [97] that provide a better understanding of the functions, regulation and complex network of sphingolipids and sphingolipid-derived mediators in human health and diseases. Further, the sphingolipids and sphingolipid metabolites signal, both, intracellularly and extracellularly via G-protein coupled receptors. All sphingolipids share a common structural feature comprising of a long-chain sphingoid base such as sphingosine [(2*S*, 3*R*, 4*E*)-2-amino octadecc-4-ene-1,3 diol]. The -NH2 (amino) group on C2 is either free or linked to long-chain (C16–C18) or very long-chain (C20–C24) fatty acids with or without double bonds to generate ceramides. Ceramides can be further derivatized by the addition of a head group such as phosphorus, phosphocholine, glucose, galactose, or several sugar residues. Small amounts of sphingoid base containing a head group such as phosphorus, phosphocholine, or sugar units without the amide-linked fatty acid termed as “lyso sphingolipids” have been identified in biological fluids and tissues. As several excellent reviews describe the chemistry and biochemistry of sphingoid bases and sphingolipids [98,99,100], only a brief outline of the metabolism of sphingolipids is presented here.

Sphingomyelin (SM), the most abundant sphingolipid in mammalian cells, is hydrolyzed by three major sphingomyelinases (SMases), namely acidic, alkaline, and neutral to generate ceramide and phosphocholine [101]. Ceramide, a pro-apoptotic lipid molecule, promotes cell cycle arrest and regulates epithelial and endothelial apoptosis in the lung tissue [102]. Ceramide can also be phosphorylated to ceramide-1-phosphate (C1P) by ceramide kinase, which in contrast to ceramide has pro-survival function and also plays a role in the inflammatory process [103]. Ceramidases (acidic, alkaline and neutral) catabolize ceramide to sphingosine, a precursor of sphingosine-1-phosphate (S1P) [104,105].

Sphingosine, derived from ceramide or dihydrosphingosine, obtained by biosynthesis from palmitoyl CoA and serine in the de novo pathway, is phosphorylated by sphingosine kinase (SPHK) 1 or 2 to generate S1P [106]. S1P is dephosphorylated to sphingosine by S1P phosphatases (SPP1 or -2, encoded by SGPP1-2) [107] or lipid phosphate phosphatases (LPPs) [107]. However, S1P can be irreversibly degraded by S1P Lyase (encoded by *SGPL1*) to ethanolamine phosphate and Δ2-hexadecenal (∆2-HDE) [106]. In addition to S1P, S1P lyase also hydrolyzes dihydro-S1P (DH-S1P) to hexadecanal and ethanolamine phosphate (Figure 2).

In cells, ∆2-HDE is subsequently oxidized to *trans*-2-hexadecenoic acid followed by CoA addition to generate *trans*-∆2-hexadecenoyl CoA, which is reduced to palmitoyl CoA by *Trans*-2-enoyl-CoA reductase (TER) in mammalian cells [108]. The palmitoyl CoA generated from ∆2-HDE is channeled to glycerophospholipids [108]; thus S1P lyase is a key enzyme in connecting the sphingolipid catabolism to glycerophospholipid metabolism. S1P is a pleotropic bioactive sphingolipid that is angiogenic and involved in several cellular functions and signals both intracellularly and extracellularly via G-protein coupled S1P_1–5_ receptors on the plasma membrane of cells [109]. Ceramide can also be converted to glucosylceramide by ceramide glucosyltransferase [110] and other glycosphingolipids [111], and to ceramide-1-phosphate catalyzed by ceramide kinase [112,113]. Ceramide-1-phosphate is a naturally occurring bioactive sphingophospholipid involved in non-receptor mediated intracellular actions of cell proliferation, migration, and inflammation [114]. It is evident that sphingolipid metabolites generated from ceramide regulate a variety of cellular functions extracellularly via G-protein coupled receptors and intracellularly by non-receptor mechanisms.

### 3.1. Sphingomyelin and Sphingomyelinase in Pulmonary Fibrosis

Sphingomyelin, once considered to be an inert but essential membrane component, is now recognized as an important bioactive lipid and precursor for ceramide and sphingosylphosphocholine in mammalian cells [115]. Sphingomyelin is hydrolyzed by sphingomyelinase (SMase), a phosphodiesterase, to ceramide, which regulates a variety of cellular processes such as apoptosis, autophagy, senescence, infection, and inflammation [104]. At least five types of SMases have been identified and classified according to their cation dependence and pH optima. The five types of SMases are: lysosomal acid SMase; secreted zinc-dependent SMase; Mg^2+^-dependent neutral SMase; Mg^2+^-independent SMase; and alkaline SMase [116]. Among the SMases, the acid SMase (ASMase) has been widely studied as it is activated by stress and specific developmental cues that result in rapid generation of ceramide in the plasma membrane. Importantly, the ASMase exists in complex with acid ceramidase that cleaves ceramide to sphingosine. ASMase and its role in several human pathologies such as Farber disease [117], and cystic fibrosis [118] are known. Recent advances in the development of small molecule inhibitors to block ASMase to reduce inflammation in preclinical models of cystic fibrosis are encouraging but unclear for IPF and further studies are necessary to explore the therapeutic potential in lung fibrosis [119,120]. ASMase plays a role in experimental PF. Mice challenged with bleomycin to induce inflammation and PF showed increased activity of ASMase and acid ceramidase in lung tissue lysates, and deletion of ASMase in mice reduced bleomycin-induced lung inflammation, collagen deposition, and development of lung fibrosis [121]. Although these data suggest a role for ASMase in bleomycin-induced PF, it is unclear if ASMase expression and activity are altered in IPF lungs and in cells from IPF lungs, which require further investigation. Additionally, *SMPD1* the gene that encodes ASMase is subject to epigenetic regulation through methylation [122], which requires further investigation in IPF.

### 3.2. Ceramide Metabolism and Signaling in Pulmonary Fibrosis

In cells, ceramide could be generated by de novo biosynthesis, sphingomyelin degradation, synthesis from sphingosine and fatty acid, degradation of glucosyl- and galactosyl-ceramide and ceramide-1-phosphate [103]; however, the SMase mediated sphingomyelin degradation probably represents the major mechanism for intracellular ceramide production by cellular stimulation and stress [123]. Ceramide is central to sphingolipid pathways involved in several human diseases and accumulation of ceramide has been shown in pathologies such as COPD, cystic fibrosis, PF, ischemia/reperfusion injury and acute inflammation. Unlike cystic fibrosis [124] and COPD [125] definitive studies relating to ceramide levels in BAL fluid, plasma, or lung tissue to collagen deposition or fibrosis in IPF lungs have not been performed. In bleomycin-induced PF, changes in ceramide levels in BAL fluid and lung tissue were not significantly different from control mice, although S1P levels were elevated [126]. However, in the radiation model of pneumonitis and fibrosis, ceramide levels in the lung tissues were decreased at one-week post-irradiation but significantly increased at six weeks, which was also seen in the BAL fluid in late stages of radiation [127]. More importantly, the ratio of ceramide to S1P levels might be a better indicator of sphingolipid involvement in the pathology. In another study, it was found that mice exposed to ^56^Fe radiation, the total lung ceramide levels were found to be elevated predominantly by the increase in palmitoyl fatty acid (C16:0)-containing ceramide molecular species [128]. While limited data are available on the sphingolipidome of lung tissues, plasma, or BAL fluid, metabolomics of IPF lungs showed dysregulated *SMPD1*, *SMPD4*, and *DEGS1* mRNA expression that point to aberrant ceramide production, whereas reduced mRNA expression of *ACER3* suggested dysregulated ceramide metabolism [129]. However, changes in mRNA levels of the dysregulated sphingolipid metabolizing enzymes were not validated by determining the protein levels in this study, as mRNA levels may not reflect true metabolomics change.

### 3.3. S1P Signaling Axis in the Pathophysiology of IPF and Animal Models of Pulmonary Fibrosis

S1P is the simplest bioactive sphingophospholipid that is present in circulating cells in the blood, biological fluids including plasma, BAL fluid, and in all the cell types in various organs. Cellular S1P levels are much lower (<0.5 µM) compared to plasma (0.5–1 µM) and tightly regulated by synthesis and degradation. About two-thirds of plasma S1P is bound to apolipoprotein M (apoM), which is a minor component of the High-density lipoprotein (HDL) particle [130,131,132]. In mammalian cells, S1P is generated by phosphorylation of sphingosine catalyzed by the lipid kinase sphingosine kinase (SPHK) 1 and 2 [133]. In addition to sphingosine, SPHK2 can also phosphorylate FTY720 a structural analog of sphingosine [134]. S1P is degraded to sphingosine by S1P phosphatases 1 and 2 [135], lipid phosphate phosphatases, and by S1P lyase to ∆2-hexadecenal and ethanolamine phosphate [109,136,137]. Platelets and erythrocytes have much higher levels of S1P as they lack S1P lyase [138]. S1P that is generated inside the cell is transported to outside by ABC transporters [139] and spinster homolog 2 (SPNS2) transporter [139]. S1P is a potent angiogenic factor and exhibits a plethora of effects on cellular functions [134] by binding to a family of G-protein coupled receptors S1P_1–5_ present on the plasma membrane of cells. S1P signals intracellularly, independent of S1P_1–5_, by binding to target proteins such as telomerase and HDACs [109], and modulates calcium homeostasis, regulates mitochondrial assembly and function by binding to prohibitin 2 [140], and is a modulator of BACE1 activity in Alzheimer’s disease [141]. Additionally, S1P has been identified as a missing cofactor required for the E3 ligase activity of TNF receptor-associated factor 2 (TRAF2) [142]. However, in keratinocytes, deletion of *Traf2*, but not *Sphk1*, disrupted TNF-α-mediated NF-kB and MAPK signaling causing skin inflammation [143]. There is overwhelming evidence for protection as well as the detrimental effects of S1P in human diseases. In lipopolysaccharide (LPS)-induced and ventilator-induced lung inflammatory injury in mice, S1P levels in plasma [144], BAL fluid, and lung tissues were significantly lower compared to controls [106], and infusion of S1P was found to be beneficial in mouse and canine models of sepsis [145,146]. However, in lung disorders such as bronchopulmonary dysplasia (BPD) [147], pulmonary arterial hypertension (PAH) [148], asthma [149], experimental models of PF [150], and IPF [151] circulating and lung tissue levels of S1P were significantly elevated compared to controls, and reducing S1P levels by genetic deletion of *Sphk1* in mice or inhibition of SPHK1 by small molecule inhibitors conferred protection. Several excellent reviews have dealt with the role of S1P signaling in sepsis [152], asthma [153], and BPD [154]; therefore in this section, the role of S1P signaling in IPF and animal models of PF involving dysregulation of S1P metabolizing enzymes will be considered.

#### 3.3.1. S1P Levels Are Altered in IPF and Animal Models of Pulmonary Fibrosis

There is only one study that describes S1P levels in IPF. S1P levels were increased in BAL fluid and serum from IPF patients [151], which correlated with lung function parameters such as diffusion capacity, forced expiratory volume, and forced vital capacity. In bleomycin- and radiation-induced mouse models of PF, sphingolipid levels were altered compared to control groups. S1P and DH-S1P levels were increased up to 4-fold in mouse lungs after 3, 7, and 14 days of post-bleomycin challenge [126,150]. The increase in S1P and DH-S1P levels in lung tissues on day 21 after bleomycin challenge were partly restored after administration of a SPHK1 inhibitor, SKI-II [126]. This decrease in S1P and DH-S1P levels correlated with protection against bleomycin-induced PF and mortality. Similarly, an increase in S1P and DH-S1P levels was detected in plasma, lung tissue, and BAL fluids 18 weeks after a 20 Gy thoracic irradiation of mice [155]. In the radiation model of PF, inhibition of sphingolipid de novo biosynthesis by targeting serine palmitoyltransferase with myriocin decreased levels of S1P and DH-S1P in mouse lung and plasma and delayed the onset of radiation-induced PF [155]. These studies show that S1P and DH-S1P levels are upregulated in human IPF and animal models of PF and blocking their production or enhancing their catabolism can be a therapeutic approach for fibrotic lung diseases.

#### 3.3.2. SPHK1/S1P Signaling Promotes Lung Inflammation and Pulmonary Fibrosis

Increased S1P levels observed in plasma, BAL fluid, and lung tissues of IPF patients and in animal models of lung fibrosis could be attributed to modulation of its metabolism mediated by SPHKs, S1P lyase, and S1P phosphatases/lipid phosphate phosphatases. SPHK1, but not SPHK2, protein expression was increased in lung tissue lysates from IPF patients compared to control subjects, as well as in the murine model of bleomycin-induced lung inflammation and PF [126,151]. Microarray analysis of peripheral blood mononuclear cells (PBMCs) for mRNA expression of S1P synthesizing enzymes (SPHK1/2) negatively correlated with DLCO (diffusing capacity of the lung for carbon monoxide) in IPF, and Kaplan-Meier survival analysis comparing IPF with control groups demonstrated significantly reduced survival of patients with high expression of *SPHK1* or *SPHK2* [126]. The causative role of SPHK1 in PF was confirmed in genetically engineered mice lacking *Sphk1*. Bleomycin upregulated the expression of SPHK1 in lungs compared to WT mice, and genetic deletion of *Sphk1*, but not *Sphk2*, attenuated bleomycin-induced mortality, lung injury and collagen deposition in the lungs. Further, administration of SPHK1 inhibitor, SKI-II to mice attenuated bleomycin-induced lung inflammation and collagen deposition in lungs confirming a role for SPHK1-mediated S1P in the development of PF [126]. The development and progression of IPF and experimental PF show the involvement of both immune- and non-immune cells. Among the several lung cell types, AECs, fibroblasts, AMs, and endothelial cells have been implicated in the pathogenesis of lung fibrosis [15,156,157]. Genetic deletion of *Sphk1* in fibroblasts and AECs, but not endothelial cells, protected mice from bleomycin-induced lung fibrosis [158]. Additionally, a role for SPHK1 was also shown using two alternative models, namely the radiation-induced lung injury/pulmonary fibrosis (RILI/PF) and asbestos-induced lung fibrosis (AIPF) models. In the RILI/PF model, the expression of both SPHK1 and SPHK2 was elevated at 6 weeks in lung tissues [127]. Further, in this model, simvastatin augmented the expression of both the isoforms of SPHK and conferred protection from radiation-induced lung injury. Although the mechanisms underlying the differential effects of simvastatin on lung SPHK1 and SPHK2 expression are unclear, evidence of these changes in RILI/PF supports the idea that SPHK1 and SPHK2 could potentially serve as useful clinical biomarkers of lung inflammatory injury. In the asbestos-induced PF model, the SPHK1 inhibitor, PF-543 mitigated collagen deposition, and development of PF in mice [159]. Thus, SPHK1 appears to be a viable target to ameliorate the development of lung fibrosis in preclinical models; however, clinical trials to determine the efficacy of SPHK1 inhibitor(s) in treating IPF are required.

#### 3.3.3. Dihydro S1P Signaling in Pulmonary Fibrosis

DH-S1P levels, similar to S1P are increased in lung tissues of bleomycin-challenged mice [150] and plasma, BAL fluid, and lung tissues of thoracic radiated mice [126,127]; however, DH-S1P levels in IPF patients have not been reported. While S1P is profibrotic and activates fibroblasts, DH-S1P seems to exhibit an antifibrotic property in fibroblasts. DH-S1P inhibited TGF-β-induced SMAD3 signaling and collagen upregulation in human foreskin fibroblasts through a PTEN/PPM1A-dependent pathway [160]. Additionally, S1P and DH-S1P showed opposing roles in the regulation of the MMP1/TIMP1 pathway in dermal fibroblasts [161,162]. DH-S1P is antifibrotic in scleroderma fibroblasts wherein PTEN protein levels were low that correlated with elevated levels of collagen and phospho-Smad3 and reduced levels of MMP1. DH-S1P treatment restored PTEN levels and normalized collagen and MMP1 expression, as well as SMAD3 phosphorylation. The distribution and function of S1P receptors differ in scleroderma and healthy fibroblasts, suggesting that alteration in sphingolipid signaling pathway may contribute to scleroderma fibrosis [163].

#### 3.3.4. S1P lyase/S1P Signaling in IPF and Pulmonary Fibrosis

S1P lyase irreversibly hydrolyzes S1P to ∆2-HDE and ethanolamine phosphate and thus regulates intracellular S1P levels [109,136]. S1P lyase plays an important role in sepsis-induced inflammatory lung injury, where circulating and lung tissue S1P levels are lower compared to controls. Inhibition of S1P lyase in vivo increased circulating S1P levels and mitigated LPS-induced lung inflammation [144] and in vitro restored LPS-induced endothelial dysfunction. In contrast to the sepsis model, inhibition of *Sgpl1* or S1P lyase activity had an opposite effect on fibrogenesis. S1P lyase expression was upregulated in IPF lung tissues, primary lung fibroblasts isolated from patients with IPF and bleomycin-challenged mice. Knockdown of S1P lyase (*Sgpl1^+/−^*) in mice augmented bleomycin-induced PF, and patients with IPF had reduced *Sgpl1* mRNA expression in PBMCs, exhibited higher severity of fibrosis and lower survival rate [164]. Thus, the sphingolipid metabolizing enzyme S1P lyase may be a potential target that would require in vivo activator(s) to reduce S1P levels in IPF patients.

#### 3.3.5. Autophagy and S1P lyase/S1P Pathway in IPF and Pulmonary Fibrosis

Autophagy, an intracellular catabolic process triggered to remove aggregated/misfolded protein(s) and damaged organelles, plays a role in IPF. Lung tissues from IPF patients demonstrated decreased autophagic activity as assessed by LC3, p62 protein expression and immunofluorescence, and numbers of autophagosomes [165]. This inhibition of autophagy was attributed to TGF-β action on fibroblasts via activation of mTORC1 and increased expression of TIGAR. However, the role of S1P and S1P signaling in autophagy is controversial. Earlier studies indicate that S1P is an inducer [166,167,168,169,170] or inhibitor of autophagy [171,172]. Studies carried out in fibroblasts isolated from *Sgpl1^+/−^* mouse lung and overexpression of h*SGPL1* in HLFs clearly established a role for S1P in bleomycin-induced autophagy. Expression of beclin1, LC3, and the total number of autophagosomes in lung fibroblasts isolated from *Sgpl1^+/−^* mice were significantly lower than in WT controls, and chloroquine treatment increased autophagosome numbers in lung fibroblasts isolated from WT mice compared with those from *Sgpl1^+/−^* [164]. Similarly, transfection of HLFs with adenoviral construct of *hSGPL1* enhanced the expression of LC3 and beclin 1, and reversed TGF-β-induced decrease of LC3 expression and autophagosome formation [164]. Similar to TGF-β signaling, the S1P-induced mRNA and protein expression of FN, α-SMA, were also suppressed by overexpression of S1P lyase in HLFs. S1P challenge also attenuated LC3 expression and autophagosome formation, and overexpression of S1P lyase blocked S1P-induced attenuation of LC3 mRNA and protein expression [164]. Thus, it has been suggested that increased expression of S1P lyase in IPF lungs could represent a compensatory mechanism to partly counterbalance the TGF-β- and S1P-induced inhibition of autophagy, and the enhanced expression of S1P lyase serves as an endogenous suppressor of PF [164]. Autophagy inhibition might also mediate epithelial-mesenchymal transition and lung myofibroblast differentiation in IPF [173]. Thus, increased S1P levels could account for EMT of AEC in IPF [151].

#### 3.3.6. Serine palmitoyltransferase Modulation of S1P Signaling and Pulmonary Fibrosis

The first and rate-limiting enzyme in the de novo biosynthesis of sphingolipids is serine palmitoyltransferase (SPT), a pyridoxal phosphate-dependent enzyme, which condenses serine and palmitoyl CoA to generate 3-ketosphinganine (3-keto dihydrosphingosine) that is subsequently converted to ceramide and sphingomyelin [174] (Figure 2). SPT is composed of two major subunits SPTLC1 and SPTLC2 that encode 53- and 63-kDa proteins, respectively [175]. Fungal metabolites such as sulfamisterin [176] and myriocin [177,178] have been employed to block SPT and interrogate the role of this enzyme on the overall sphingolipid metabolism in animals and pathologies. In a major study by Gorshkova et al., a single dose of myriocin decreased radiation-induced pulmonary inflammation and fibrosis and alleviated the dysregulated lung gene expression at 18 weeks post-radiation [155]. Additionally, myriocin inhibited the upregulation of S1P/DH-S1P levels and modified ceramide-sphingoid base molecular species levels in the irradiated animals. Myriocin also modulated the expression and/or activity of S1P metabolizing enzymes in the lung tissue. Myriocin treatment attenuated the radiation-induced increase in expression of SPHK1, SPT, and SPGL1, but not SPHK2; however, the activity of S1P lyase was not modulated compared to control animals. Myriocin elicited its effect by attenuating TGF-β-induced α-SMA and SPT2 expression and myofibroblast differentiation in HLFs [155]. Thus, the ability of myriocin to ameliorate radiation-induced lung inflammation and fibrosis suggests that SPT might be a novel therapeutic target in radiation-induced lung fibrosis.

#### 3.3.7. S1P Receptors in Pulmonary Fibrosis

S1P signals intracellularly and extracellularly via five G-protein coupled receptors S1P_1–5_. Additionally, S1P generated in the nucleus signals within the nucleus independent of S1P_1–5_ and is involved in epigenetic regulation of pro-inflammatory genes activated by *Pseudomonas aeruginosa* in the lung epithelium [155]. The S1P_1–5_ are coupled to G_i_, G_α_, G_o_, G_q_, and G_12/13_ and this differential coupling dictates S1P signaling, in part, to various downstream effectors such as MAPKs, PI3K, Src, nMLCK, adenylate cyclase, phospholipase C (PLC), phospholipase D that result in a plethora of cellular responses [134,137]. A growing body of evidence suggests a role for S1P_2,3_ in PF; however, the role of S1P_1_, the predominant S1P receptor expressed in many mammalian cells, is unclear as complete deletion of *S1p1* is embryonically lethal. However, some light has been shed on the S1P_1_ role in a vascular leak/lung fibrosis. In the bleomycin murine model, prolonged exposure of FTY720 (a non-selective S1P_1,3_ modulator) and AUY954 (an S1P_1_ selective modulator) caused a pulmonary leak in mouse lungs while low doses of bleomycin did not induce lung fibrosis. However, administration of either FTY720 or AUY954 along with low doses of bleomycin exacerbated vascular leak that was accompanied by intra-alveolar coagulation and development of extensive lung fibrosis in mice [179]. To understand the mechanism of FTY720 or AUY954 + bleomycin effect on the development of lung fibrosis in the context of a vascular leak, an in vivo thrombin coagulation-vascular leak model was investigated in the presence of bleomycin. Inhibition of thrombin-induced coagulopathy with an anticoagulant dabigatran, but not warfarin, attenuated lung fibrosis mediated by FTY720+low doses of bleomycin [180]. The FTY720 + bleomycin-induced vascular leak correlated with increased α_v_β_6_ expression in the lung and thrombin inhibition with dabigatran diminished α_v_β_6_ expression and activation of the TGF-β canonical signaling [180]. However, this in vivo FTY720 + bleomycin model does not explain the ability of FTY720 to enhance endothelial barrier function independent of S1P_1_ in lung endothelial cells [181]. In contrast to FTY720, another analog, FTY720 (*S*)-phosphonate, that is non-hydrolyzable by lipid phosphate phosphatases, significantly inhibited bleomycin-induced alveolar-capillary leakage and inflammatory cell recruitment while FTY720 failed to confer protection against bleomycin-mediated lung inflammatory injury [182]. The mechanism of protection by FTY720 (*S*)-phosphonate is unclear, but it preserved expression of S1P_1_ on the cell surface while FTY720 allowed S1P_1_ internalization and recycling in endothelial cells.

S1P_2_ and S1P_3_ have been shown to be pro-inflammatory and profibrotic in the bleomycin model of lung fibrosis. Genetic deletion of *S1pr2* or pharmacological inhibition of S1PR2 alleviated bleomycin-induced PF [183]. Bone marrow chimera experiments showed that bone marrow-derived cells contributed to the development of lung fibrosis, and depletion of macrophages also reduced bleomycin-induced lung fibrosis. Further, bleomycin challenge increased expression of pro-inflammatory cytokines such as IL-4 and IL-13, which were diminished in *S1pr2-*deleted mice [183]. A similar role for *S1pr2* in the development of bleomycin-induced lung fibrosis was demonstrated using *S1pr2* KO mice and JTE-013 [184], a pharmacological inhibitor of S1PR2 in mice [184]. Knockdown of *S1pr3* attenuated bleomycin-induced lung inflammation and PF in mice without changing TGF-β levels, but by reducing connect tissue growth factor (CTGF) [185]. S1PR1 and S1PR3 agonists such as FTY720-phosphate and Ponesimod, but not SEW2871, caused a robust stimulation of ECM synthesis and expression of pro-fibrotic genes including CTGF [186]. Depletion of *S1PR2*, or *S1PR3*, but not *S1PR1*, in HLFs attenuated Rho activity that is closely associated with fibrosis and differentiation of myofibroblasts [187]. Thus, both *S1PR1* and *S1PR3* could be therapeutic targets in PF. Interestingly, Fingolimod (FTY720) at 1.25 mg and 5.0 mg daily dose given to patients with relapsing multiple sclerosis showed a reduction in pulmonary function [188] as observed in IPF patients.

#### 3.3.8. Interaction between TGF-β and S1P Signaling in the Pathogenesis of Pulmonary Fibrosis

TGF-β is a critical cytokine that drives the development of IPF and PF in animal models. TGF-β promotes EMT in AECs [189,190], fibroblast to myofibroblast transdifferentiation [191], and EndMT, which are key pathways involved in the development of lung fibrosis. TGF-β expression is increased in the lungs of IPF patients and preclinical models of lung fibrosis [17,192,193]. TGF-β binds to TGF-βRII and initiates binding to and phosphorylation of TGF-βRI. This triggers recruitment of SMAD2/3 to the cytoplasmic domain of TGF-βRI, phosphorylates SMAD2/3, SMAD2/3 forms a trimer with SMAD4, which translocates to the nucleus where it binds to SMAD-binding elements in promoter regions to modulate gene transcription [194]. Conditional deletion of TGF-βRII in lung epithelial cells protected mice from bleomycin-induced fibrosis, but these mice developed emphysema-like phenotypes [195,196]. Similarly, *Smad3-*deficient mice showed alveolar destruction resembling emphysema but developed lung fibrosis in response to bleomycin [197]. In addition to the canonical pathway of signaling via SMAD, TGF-β can signal via the non-canonical pathways such as MAPKs, PI3K/Akt, and Rho-GTPase to modulate AECs and fibroblasts to stimulate a fibrotic phenotype [18,198]. Interactions between TGF-β/TGF-βR signaling with serotonin, integrins, sGC-cGMP-PKG, S1P/S1PRs, and lysophosphatidic acid (LPA)/ LPA receptors (LPARs) have been demonstrated. There is overwhelming evidence indicating the importance of SPHK1/S1P signaling in TGF-β-mediated fibroblast to myofibroblast differentiation. In HLFs, downregulation of *Sphk1* with siRNA or inhibition of SPHK1 activity with inhibitors decreased α-SMA and fibronectin expression upregulated by TGF-β [126,150,187]. Further, knocking down S1P_2_ and S1P_3_, but not S1P_1_, with siRNA reduced TGF-β-induced α-SMA expression, and blocking SPHK1 had no effect on SMAD2/3 phosphorylation [187,199]. In the murine model of lung fibrosis, bleomycin-induced TGF-β secretion and phosphorylation of SMAD2/3, AKT, JNK1, and p38 MAPK were mitigated in *Sphk1* deleted or SPHK1 inhibitor administered mice, demonstrating the interaction between the TGF-β and SPHK1/S1P signaling axis in vivo [126,150]. Both in vivo and in vitro, TGF-β upregulated SPHK1 and enhanced S1P levels and exogenous addition of S1P antibody to fibroblasts prevented α-SMA and fibronectin increase [126,150], further confirming the release of S1P from the cell for its extracellular responses. Similarly, TGF-β-induced EMT in A549 epithelial cells was partially dependent on SPHK1/S1P_2,3_ signaling axis [151]. TGF-β increased the expression of both SPHK1 and S1P lyase in HLF, which was blocked by anti-TGF-β neutralizing antibody and *SMAD3* siRNA [164]. The transcriptional activity of *SMAD3* in regulating *SGPL1* expression induced by TGF-β was verified by ChIP assay as well as *h**SGPL1* luciferase promoter activity in HLF [164]. The transcriptional regulation of *SPHK1* by SMAD3 or other transcriptional factors in response to TGF-β in normal and IPF lung fibroblasts needs to be further investigated. The TGF-β-mediated fibroblast differentiation to myofibroblast was attenuated by overexpression of *SGPL1* in HLF and is attributed to the decrease in intracellular S1P level, increase in expression of LC3 and Beclin1, and autophagy [164]. Thus, understanding the balance in expression of SPHK1 and S1P lyase in lung tissues of IPF and animal models will provide a handle to S1P signaling in the progression or resolution of PF (Figure 3).

#### 3.3.9. SPHK1/S1P Signaling and Mitochondrial ROS in Pulmonary Fibrosis

The pathophysiology of IPF is characterized by increased ROS production and an imbalance in redox status of the lung tissue [200,201]. The two major sources of ROS in cells are from mitochondrial oxidative phosphorylation and activation of NADPH oxidases (NOXs) [202,203]. Altered mitochondrial homeostasis that includes increased mitochondrial ROS (mtROS), impaired respiration, mtDNA damage, compromised mitochondrial dynamics and mitophagy have been reported in epithelial cells and fibroblasts from healthy aged lungs, IPF lungs and lungs from murine models of bleomycin- and asbestos-induced PF, resulting from upregulation of NADPH Oxidase (NOX) 4, and increased mtROS and mtDNA damage [204,205,206,207]. TGF-β has been shown to induce mtROS through decreased Complex IV activity in senescent cells [208,209], and recent studies suggest that SPHK1/S1P signaling stimulates NOX2-dependent ROS generation in lung endothelial cells [137,210,211,212], and mtROS in lung fibroblasts [158]. TGF-β stimulated mtROS in HLF that was dependent on SPHK1 expression and activity as well as YAP activation. Inhibition or downregulation of SPHK1, and YAP1 activity or expression reduced TGF-β mediated mtROS and scavenging mtROS with MitoTEMPO attenuated TGF-β-dependent expression of fibronectin and α-SMA, demonstrating a definitive role for mtROS in fibroblast differentiation. Further, genetic deletion of Sphk1 in mouse lung fibroblasts or inhibition of SPHK1 with PF543 (a specific inhibitor of SPHK1) reduced bleomycin-induced YAP1 co-localization with FSP-1 in fibrotic foci rich in fibroblasts [158]. This study demonstrated a role for SPHK1/S1P signaling in TGF-β-induced YAP1 activation that is essential for mtROS generation and expression of fibronectin and α-SMA in fibroblasts. How SPHK1 activates YAP1 in HLFs is unclear. It is important to determine if both SPHK1 and YAP1 are translocated to the mitochondrial outer membrane to initiate the process of mtROS production in response to a stimulus such as TGF-β. In confluent cells, YAP1 is primarily phosphorylated and localized in the cytosol, and upon activation gets dephosphorylated by phosphatases including PTPN14 [213] and translocates to the nucleus and functions as a co-transcriptional regulator with TAZ [214,215] (Figure 3).

## 4. Phospholipids and Phospholipid Metabolizing Enzymes in Pulmonary Fibrosis

Phospholipids are essential components of all biological membranes and metabolic dysregulation of phospholipids have been shown to contribute to the pathogenesis of several pulmonary disorders including IPF and experimental PF. Lipidomics of IPF lungs or lungs of animal models have not been performed; however, phospholipid content in the BAL fluids was reduced in IPF lungs and mouse lungs of animal models. Aberrant phospholipid metabolism by phospholipid metabolizing enzymes such as PLA_2_, PLC, PLD, and lyso PLD or autotaxin (ATX) generate bioactive lipid molecules that play key roles in the development and progression of lung fibrosis [216]. Further, metabolism of cardiolipin, a major mitochondrial phospholipid, also seems to contribute to the pathogenesis of IPF. The role of arachidonic acid-derived prostanoids and leukotrienes on lung fibrosis have been dealt with in the earlier section and here we will review the recent developments on fatty acid, PLD/PA, ATX/LPA, cardiolipin metabolism by lysocardiolipin acyltransferase (LYCAT) and oxidized phospholipids in the pathophysiology of lung fibrosis.

### 4.1. Surfactant Lipids and Surfactant Proteins in IPF and Pulmonary Fibrosis

The lung surfactants, in addition to dipalmitoyl phosphatidylcholine, dipalmitoyl phosphatidylglycerol and phosphatidylinositol also include four surfactant proteins, SP-A, SP-B, SP-C and SP-D, which act as a defense mechanism against toxic pathogens or microbes trying to invade the lung. SP-A and SP-D serve as biomarkers of IPF, as seen by their ability to predict the survival rates in IPF patients. In the BAL fluid, the levels of phosphatidylglycerol were lower while the contents of phosphatidylinositol and sphingomyelin were elevated, and the dysregulated phospholipid levels correlated with the severity of the disease [217,218,219,220]. Administration of a natural bovine lung extract neutral lipids, phospholipids enriched with phosphatidylethanolamine, fatty acids, and surfactant proteins attenuated bleomycin-induced lung fibrosis and soluble collagen levels in mouse lungs [221]. This protective role of the administered surfactant was attributed to the inclusion of phosphatidylethanolamine in the surfactant preparation. SP-D levels were found to be elevated in the serum of patients with radiographic abnormalities during IPF [222], whereas elevated levels of SP-A and SP-D were seen in IPF patients and also systemic sclerosis patients when compared to normal individuals [223]. Interestingly, another study showed that SP-D levels were increased only in patients who were diagnosed in late stages of IPF and could be used as a biomarker for progression of disease during anti-fibrotic treatments [222]. SP-B levels were increased in patients with IPF [224,225], and BAL fluid analysis showed that SP-A/phospholipid ratio in the BALF was lower in IPF patients and it could be used a predictive marker for survival in these patients [225]. Mutations in SP-C were seen in familial and sporadic cases of PF. The levels of SP-B were decreased, and SP-D was increased during bleomycin-induced fibrosis in the mice, but there was no change in the levels of SP-A and SP-C [226]. SP-C deficient mice were found to have increased infiltration of inflammatory cells, lung architectural distortion, and increased collagen deposition. There was also delayed resolution of fibrosis in these mice as seen by sustained apoptosis of the lung parenchymal cells and prominent fibrosis in the centriacinar and sub-pleural regions containing fibroblasts, collagen fibrils, and damaged interalveolar septa. Mitochondrial fusion related proteins such as Mitofusin 1 (Mfn1) and Mitofusin 2 (Mfn2) regulate surfactant production in alveolar type II epithelial cells by playing a key role in lipid metabolism which in turn regulates fibrosis in the lungs. Loss of Mfn1 and Mfn2 in AEC in mice was found to promote bleomycin-induced lung fibrosis in a recent study [227]. More interestingly, fibrosis was spontaneously induced in mice with both, Mfn1 and Mfn2 deletion in AECs. AEC cells from *Mfn1^−/−^* and *Mfn2^−/−^* mice challenged with bleomycin were found to have altered lipid metabolism of cholesterol, ceramides, phosphatidic acids, phosphatidylcholine, phosphatidylethanolamine, phosphatidylserine, plasmalogen phosphatidylethanolamine, and decreased SP-B, SP-C gene expression [227]. Mfn1/2-deficient AECs showed no changes in the SP-B, SP-C gene expression, but with altered lipid profile by changes in mono-acylglycerol, diacylglycerol, acylcarnitine, cholesterol, phosphatidylserine, and phosphatidylglycerol levels. Thus, mitochondrial dynamics seems to have an impact on the altered lipid metabolism in AECs due to bleomycin-induced lung injury.

### 4.2. Phospholipase D/Phosphatidic acid Signaling Axis in Development of Pulmonary Fibrosis

PLD hydrolyzes phosphatidylcholine to phosphatidic acid (PA), and choline [228,229,230,231]. It can degrade other phospholipids such as phosphatidylethanolamine and phosphatidylserine to PA and ethanolamine or serine, respectively. In addition to the phosphohydrolase activity, it has a transphosphatidylation activity where the PA is transferred to primary short-chain alcohols such as methanol, ethanol, butanol, and propanol, but not secondary or tertiary alcohols, generating the corresponding phosphatidylalcohol [230]. There are six isoforms of PLD, PLD1-6, of which PLD1 and PLD2 isoforms exhibit the ability to hydrolyze phospholipids and these two isoforms have been widely recognized in several human pathophysiologies including cancer, hypertension, neurodisorders, diabetes, and acute lung injury [228,229,230,231]. PA is a bioactive lipid second messenger, which is further converted to diacylglycerol (DAG) or lyso-PA (LPA) by PA phosphatase [232,233] or PA-specific PLA_1_/PLA_2_ [234,235], respectively. PLD mediated PA generation is involved in the regulation of various cellular processes including cell survival, cell migration, cell proliferation, differentiation, cytoskeletal changes, membrane trafficking, and autophagy [230,236,237]. PLD was activated by bleomycin in lung endothelial cells and led to reactive oxygen species generation [238,239]. PLD activation and PA generation induced by bleomycin in bovine lung ECs were significantly attenuated by the thiol protectant (*N*-acetyl-l-cysteine), antioxidants, and iron chelators suggesting the role of ROS, lipid peroxidation, and iron in the process. This study revealed a novel mechanism of the bleomycin-induced redox-sensitive activation of PLD that led to the generation of PA, which was cytotoxic to lung ECs, thus suggesting a possible bioactive lipid-signaling mechanism of microvascular disorders encountered in PF [238]. A more recent study showed that PLD2, but not PLD1, expression was elevated in lung tissues of IPF patients compared to control subjects and in lung tissues of mice with bleomycin-induced PF [239]. Both *Pld1**^−/−^*and *Pld2**^−/−^* deficient mice were protected against bleomycin-induced lung inflammation and fibrosis, thereby establishing the role of PLD in fibrogenesis. Further, bleomycin stimulated mitochondrial superoxide production, mtDNA damage, and apoptosis, which was attenuated by the catalytically inactive mutants of PLD1 or PLD2, downregulation of PLD2 expression with siRNA or inhibition of PLD1 and PLD2 with specific small molecule inhibitors [239]. The downstream targets of PLD/PA signaling in the activation of mtROS and fibroblasts are unclear; however, PA is known to stimulate SPHK1 [239], IQGAP1 via RAC1 [240], mTOR [241], and modulates mitochondrial function and dynamics [242]. In addition to PLD1 and PLD2, a novel role for PLD4 in the development of kidney fibrosis has been reported. Genetic deletion of PLD4, a transmembrane glycoprotein and an isoform of PLD that lacks any enzyme activity, in kidney tubular epithelial cells attenuated development of kidney fibrosis [243]; however, the mechanism of protection in the absence of PA generation is unknown.

### 4.3. Diacylglycerol Kinase in Radiation-Induced Fibrosis

Diacylglycerols are lipid intermediates generated de novo in cells for the biosynthesis of triglycerides (TGs) and phospholipids and a product of PLC and PLD pathways. DAGs are physiological and endogenous activators of protein kinase C (PKC) conventional (α, β, γ) and novel (δ, ε, θ, η) isoforms that are known regulators of pleiotropic downstream signaling cascades in cells. DAGs do not accumulate and are converted to PA by DAG kinase (DAGK) or TGs by acyltransferases. Of the ten isoforms of DAGK, DAGK-α has been shown to promote radiation-induced fibrosis [244]. Radiation-induced transcription of DAGK-α in cells was facilitated by profibrotic transcription factor early growth response 1 and DNA methylation profiling showed a hypomethylation pattern [245]. In dermal fibroblasts isolated from breast cancer patients, DAGK-α regulated TGF-β-induced profibrotic activation and lipid signaling as evidenced by siRNA downregulation of the enzyme or inhibition by R59949 [246]. In addition, activation of Collagen 1A mRNA expression in fibroblasts by ionizing radiation or bleomycin was attenuated by DAGK-α siRNA or the inhibitor, R59949. Inhibition of DAGK-α increased the accumulation of 7 molecular species of DAG and reduced the accumulation of specific PA and LPA molecular species [245]. In contrast to the radiation model of fibrosis, a 4-fold upregulation of C18:1/C24:2 DAG species was observed in lung tissues from bleomycin-challenged mice; however DAGK-α expression and levels of PA or LPA were not determined and the VEGF inhibitor CBO-P11 had no effect on the DAG levels due to bleomycin treatment [247]. Thus, DAGK-α is an epigenetically regulated lipid kinase involved in radiation-induced fibrosis and may serve as a marker and therapeutic target in radiation therapy.

## 5. Lysophospholipids and Lysophospholipids Metabolizing Enzymes in Pulmonary Fibrosis

Lysophospholipids such as lysophosphatidylcholine (LPC) and lysophosphatidylethanolamine (LPE) are generated from PC and PE, respectively by PLA_1_/PLA_2_ while 1-acyl or 2-acyl lysophosphatidic acid (LPA) is formed from 1-acyl- or 2-acyl LPC by the action of lyso PLD or ATX [248] (Figure 4).

Lysophospholipids are not major components of normal cellular lipids but their levels are increased in several human pathologies. LPC levels were elevated in serum samples from IPF patients as determined by ultra-high-performance liquid chromatography coupled to high-resolution mass spectrometry, and it was suggested that LPC could serve as a biomarker for IPF [249]. However, metabolomics analysis of peripheral blood samples from abnormal interstitial lung patients revealed downregulation of 1-acyl LPC but upregulation of PC, PA, and PE suggesting potential activation of lipid metabolizing enzymes during the development of interstitial lung abnormalities [250]. However, LPA, the major lysolipid linked to development of IPF, was not identified in the analysis. Interestingly, LPA levels were significantly elevated in exhaled breath condensate (EBC) of IPF patients compared to normal subjects with LPA 22:4 as the predominant molecular species [251]. In contrast to EBC, plasma LPA levels were not significantly different between the IPF and normal subjects. LPC levels were not measured in EBCs in this study. LPA levels were also elevated in BAL fluids from segmental allergen-challenged asthmatics compared to control subjects with specific upregulation of LPA 22:5 and LPA 22:6 molecular species [252]. The presence of these unusual polyunsaturated LPA molecular species in EBC, and BAL fluids, but not in plasma, suggests that these could serve as potential biomarkers in IPF and other lung diseases. Cardiolipin is metabolized by mitochondrial PLA_2_ to mono- and dilyso-cardiolipin during normal and oxidative stress and the lysocardiolipin is converted back to cardiolipin by two remodeling enzymes, tafazzin [253], and LYCAT [254]. Mutations in the X-linked tafazzin gene result in the accumulation of mono-lysocardiolipin and development of Barth syndrome [255]. LYCAT plays an important role in IPF and bleomycin-induced PF in mice [254].

### 5.1. Lysocardiolipin Acyltransferase in Cardiolipin Remodeling

Cardiolipin (CL), a major phospholipid of mitochondria, plays an important role in the structural organization, energy metabolism, and functioning of the mitochondria [256,257]. It is located mainly in the inner mitochondrial membrane (IMM), where it interacts with a number of mitochondrial proteins and enzymes. CL has been identified as an integral component of mitochondrial electron transport complex III, IV, and the ADP/ATP carrier and is essential for the stability of the quaternary protein structure. There is evidence that changes in composition and distribution of CL molecular species may be involved in the impairment of oxidative phosphorylation [258,259]. Conversion of CL to monolyso CL (MLCL) by mitochondrial phospholipase A2 has been implicated in the process of apoptosis through its interaction with a number of death-inducing proteins including cytochrome c, t-Bid, and caspase-8 [260,261,262]. Recent studies suggest the importance of CL in the severity of lung injury in experimental pneumonia [263] and role of CL fatty acid composition in mitochondrial cell function [258]. LYCAT is a key enzyme involved in the remodeling of mitochondrial CL from monounsaturated to polyunsaturated fatty acyl chains (>60% linoleic acid (C18:2) levels), and LYCAT transfers poly-unsaturated fatty acyl CoAs to mono- and dilyso-CL resulting in mitochondrial CL remodeling [254,258](Figure 5).

Bleomycin challenge significantly decreased the cardiolipin level and mol% of C18:1 and C18:2 fatty acids in mouse lung compared to the control group. In contrast to C18:1 and C18:2, the mol% of C18:0 was higher in the lung of bleomycin-challenged mice. Bleomycin challenge shifted the unsaturated to saturated fatty acid ratio and unsaturation index were lower in bleomycin-challenged animals, indicating loss of unsaturated fatty acids, especially C18:1 and C18:2 fatty acids in the cardiolipin. Overexpression of *hLYCAT* in bleomycin-treated mouse lung restored the cardiolipin level and unsaturated to saturated fatty acid ratio and unsaturation index [254].

#### Lysocardiolipin Acyltransferase and Pulmonary Fibrosis

LYCAT or Acyl-CoA cardiolipin acyltransferase (ALCAT1) is one of the key enzymes involved in remodeling the fatty acid composition of cardiolipin from saturated and monounsaturated to polyunsaturated (~60% linoleic acid) fatty acids in the mitochondria [264]. Defective cardiolipin remodeling and loss of linoleic acid cause dilated cardiomyopathy and Barth syndrome, a genetic disorder characterized by mitochondrial dysfunction, growth retardation, and neutropenia [265]. LYCAT mRNA expression was significantly reduced in PBMCs of IPF patients; however, in lung tissues from patients with IPF, and in two preclinical murine models of IPF, bleomycin- and radiation-induced PF, the LYCAT protein expression was significantly higher compared to controls. LYCAT mRNA expression in PBMCs directly and significantly correlated with carbon monoxide diffusion capacity. In murine models, *hLYCAT* overexpression reduced several indices of lung fibrosis induced by bleomycin and radiation, whereas downregulation of native LYCAT expression by siRNA accentuated fibrogenesis in the preclinical bleomycin- and radiation mice models. In vitro, LYCAT overexpression attenuated bleomycin-induced cardiolipin remodeling, mitochondrial membrane potential, ROS generation, and apoptosis of AECs. Thus, modulating LYCAT expression could offer a novel approach to ameliorate the progression of lung fibrosis.

LYCAT expressed in AECs of IPF lungs [254] and in isolated fibroblasts from fibrotic IPF lung specimens [266]. TGF-β stimulated LYCAT expression in HLF, which was dependent on SMAD3 and mutation of the SMAD2/3 binding sites (−179/−183 and−540/−544) reduced TGF-β-stimulated LYCAT promoter activity. Overexpression of LYCAT attenuated TGF-β-induced mitochondrial and intracellular oxidative stress, NOX4 expression, and differentiation of HLFs. MitoTEMPO, a mitochondrial ROS scavenger, blocked TGF-β-induced mitochondrial superoxide, NOX4 expression, and differentiation of HLFs. Treatment of HLF with NOX1/NOX4 inhibitor, GKT137831, also attenuated TGF-β induced fibroblast differentiation and mitochondrial oxidative stress. These results suggest that TGF-β stimulates LYCAT expression that negatively regulates TGF-β-induced lung fibroblast differentiation by modulation of mitochondrial ROS. Thus, TGF-β could function as a pro-fibrotic and anti-fibrotic cytokine in lung fibroblasts and fibrogenesis.

### 5.2. Autotaxin (Lysophospholipase D), LPA and LPARs

LPA is the simplest naturally occurring glycerophospholipid and consists of a glycerol backbone attached to a long-chain fatty acid of varying chain length and saturation/unsaturation and a free polar phosphate group. LPA is found in all cells and biological fluids including plasma, serum, and BAL, and its levels in biological fluids and tissues are elevated in several human pathologies including IPF and other forms of PF [248,252,267,268,269]. Plasma levels of LPA (~0.1–1.0 µM) are much lower compared to serum (>1.0 µM) due to release of LPC, a precursor for LPA production, from activated platelets and other circulating cells [248,270].

#### 5.2.1. LPA Production in Cells by Autotaxin, Phospholipase D, and Acylglycerol Kinase

At least two major pathways have been identified for LPA production in biological systems. The first pathway is mediated by autotaxin (ATX) or lysophospholipase D that utilizes extracellular LPC as a substrate to generate LPA and the majority of plasma LPA is produced via this mechanism [248,269,271]. Intra- and extra-cellular LPA levels are also regulated by membrane-associated lipid phosphate phosphatases [107,272]. ATX is an ectonucleotide pyrophaphatase-phosphodiesterase 2 (ENPP2) and encoded by *ENPP2* gene in mammalian cells [273,274]. It is secreted as an active protein from cells and is not a transmembrane protein such as other ENPPs [275]. It was later discovered that plasma ATX has lyso PLD activity [276] and uses LPC or lysophosphatidylserine as the substrate [248,271]. LPC is abundant in plasma and is associated with albumin and lipoproteins [248,277,278]. LPC is generated through the hydrolysis of PC by PLA_1_ or PLA_2_ and lecithin cholesterol acyltransferase (LCAT) enzyme. LCAT is a transacylase that transfers fatty acid from *sn*-2 position of PC to cholesterol to generate LPC and cholesteryl ester that is predominantly C81:1 (oleic acid) [248]. ATX can also utilize sphingosylphosphorylcholine as a substrate to generate S1P but the physiological relevance of this pathway is unclear. ATX is secreted by macrophages, AECs, and adipocytes, and increased ATX expression and activity has been reported in various human pathologies and experimental models mimicking human diseases [252,267,271].

The second pathway of intracellular LPA generation in cells involves phospholipase A1/A2 mediated hydrolysis of PA derived from either the de novo biosynthesis or PLD1/PLD2 signaling axis that uses PC, PS, and PE as substrates. While it will be difficult to distinguish between the two pools of PA generated inside the cell, specific PLA_1_ or PLA_2_ can hydrolyze PA to 1-acyl or 2-acyl-*sn*-glycero-3-phosphate within the cell [234,235]. The fatty acid composition of LPA derived from ATX/LPC vs. PLA_1_-PLA_2_/PA pathways may vary based on the fatty acid composition of the substrate. The third minor pathway of intracellular LPA production involves phosphorylation of 1-acyl- or 2-acyl-monoglycerol by acylglycerol kinase (AGK) [279,280]. AGK is a mitochondrial lipid kinase and PA generated in the mitochondria by AGK can serve as a substrate for the biosynthesis of CL in the mitochondria. Further, AGK is a subunit of the mitochondrial TIM22 protein import complex. Mutations in *AGK*, independent of its kinase activity, dysregulates mitochondrial protein import leading to Sengers syndrome [281]. The role of AGK in lung pathologies has not been investigated.

#### 5.2.2. LPA Signals via LPARs

The numerous physiological and pathophysiological responses of LPA are mediated through six 7-transmembrane G-protein coupled receptors (GPCRs) collectively termed as LPA_1–6_. The six receptors are widely distributed with overlapping specificities and different affinity for LPA [282]. The LPARs are coupled differentially to G_i_, G_12/13_, G_q/11_, and G_s_ that initiate numerous signaling cascades within the cell. Several studies have identified stimulation of MAPKs, PI3K, Rho, Rac, and phospholipase activation by LPA through differential coupling to various G_α_ proteins. Of the six LPARs, LPA_6_ has sequence homology with P2Y receptor and higher affinity to 2-acyl LPA than 1-acyl LPA [283]. These receptors are highly expressed in lung cells and some of the LPARs are actively involved in the development and progression of several lung pathologies.

#### 5.2.3. PPARγ is an Intracellular Receptor of LPA

The transcriptional factor, PPARγ regulates a wide range of physiological and pathophysiological activities including energy metabolism, inflammation, atherogenesis, and fibrosis [284,285]. LPA and 1-alkyl LPA (alkyl group instead of acyl group) have been shown to be agonists for PPARγ in mammalian cells [286]. PPARγ is also activated by oxidized phospholipids, fatty acids, eicosanoids, and oxidized-LDL [287] and activation of PPARγ inhibits the differentiation of fibroblast to myofibroblast [32]. In contrast to LPA, cyclic PA (cPA), the structural analog of LPA, is generated by PLD2 mediated hydrolysis of 1-acyl LPC [288] and has been shown to be a physiological inhibitor of PPARγ [289,290]. The pathophysiological role of LPA and cPA in modulation of PPARγ and development of lung fibrosis is unclear.

#### 5.2.4. ATX/LPA Signaling Axis in Pulmonary Fibrosis

Elevated LPA levels were found in BAL fluids of IPF patients and the bleomycin murine model [268,291]. The increased LPA levels in BAL fluid from IPF patients mediated fibroblast recruitment and vascular leak [268]. LPA stimulated migration of fibroblasts [292] and induced expression of TGF-β, fibronectin, α-SMA, and collagen via AKT, SMAD3, and MAPK pathways in HLF [293]. LPA signaling via LPARs is known to transactivate PDGFR via PLD2 [294] and EGFR [295] in primary human bronchial epithelial cells. However, in fibroblast activation, LPA increased TGF-β expression that signaled via TGF-βRI/TGF-βRII and this was blocked by anti-TGF-β antibody. This suggests the crosstalk between LPA signaling and TGF-β signaling in fibroblast differentiation.

*ENPP2* was identified as a candidate gene regulating lung formation, development, and remodeling by genome-wide linkage analysis coupled with expression profiling [296]. Based on this, the deletion of both alleles of *ENPP2* in mouse (*ENPP2^−/−^*) was embryonically lethal [297,298,299]; however, knockdown of a single allele (*ENPP2^+/−^*) in mouse had no major phenotypic effect in lungs. Overexpression of *ENPP2* in bronchial epithelium or liver resulted in a 2-fold increase of ATX expression in plasma without any phenotype suggesting ATX/LPA signaling axis per se has no consequence on lung morphology and function [300]. ATX is constitutively expressed in bronchial epithelium, AMs and other cells in both humans and rodents, and increased ATX staining mainly localized in bronchial epithelial cells around fibroblastic foci was seen in lung tissues from IPF patients [291,301]. The upregulation of ATX directly correlated with disease progression and irreversible PF development. A similar increase in ATX staining and protein expression was observed in the lungs of bleomycin-challenged mice. Conditional deletion of *ENPP2* in bronchial epithelial cells and macrophages reduced ATX levels in BAL fluid and disease severity confirming a pathophysiological role for ATX in lung fibrosis [291]. A number of small molecular weight inhibitors have been developed to ameliorate IPF and reverse fibrosis in patients. ATX inhibitors GWJ-A-23 [291], PF-8380 [216], BBT-877 [302], and GLPG1690 [303] have been shown to confer protection against bleomycin-induced PF in mice. Among the various ATX inhibitors, the Galapagos compound GLPG1690 has advanced to Phase III clinical trials and awaiting outcome on safety and side effects. The ATX inhibitor PAT-048 (Bristol Myers Squibb) seems to have a different mechanism in attenuating PF mediated by bleomycin [301]. PAT-048 only blocked ATX-dependent LPA production in plasma and had no effect on bleomycin-induced lung fibrosis and BAL fluid LPA levels. Additionally, there was discordance in LPC and LPA molecular species from plasma and BAL fluid. In the BAL fluid, the predominant LPA molecular species were long-chain polyunsaturated C22:5 and C22:6 fatty acids whereas the BAL fluid LPC molecular species contained shorter and saturated C16:0 and C18:0 fatty acids. It was concluded that an alternate ATX-independent pathway(s) was most likely contributed for the local generation of highly polyunsaturated LPA species in the bleomycin-injured [301]. Lipidomic analysis revealed the presence of highly unsaturated fatty acids in phospholipids of lung bronchial epithelial cells, alveolar type II epithelial cells, and AMs [304], indicating the phospholipids in lung cells could be a substrate for PLD to generate PA with polyunsaturated fatty acids that could generate LPA in the lung.

#### 5.2.5. LPA Signaling via LPA Receptors in the Development of IPF and Pulmonary Fibrosis

LPA signaling via LPA_1–3_ has been identified to be involved in the development of PF. Fibroblasts isolated from IPF lungs had elevated expression of LPA_1,_ and inhibition of LPA_1_ reduced fibroblast responses to chemotactic activity [268]. The number of apoptotic cells present in alveolar and bronchial epithelia was significantly reduced in *Lpar1* deficient mice that were exposed to bleomycin, and LPA signaling through LPA_1_ induced apoptosis of normal bronchial epithelial cells [305]. In contrast to epithelial cells, LPA signaling through LPA_1_ promoted resistance of lung fibroblasts to apoptosis. An antagonist of LPA_1_, BMS-986020, was found to lessen the extent of the decline in the forced vital capacity of lungs in IPF patients [306] while another LPA_1_ antagonist AM966 attenuated bleomycin-induced vascular leakage, lung injury, inflammation and fibrosis [307]. Thus, LPA signaling via LPA_1_ promotes apoptotic and profibrotic responses in lung epithelial cells and fibroblasts, respectively. Other LPA_1_ antagonists such as AM152, SAR100842 and ONO-7300243 have been shown to be beneficial in vivo against bleomycin mediated lung inflammation and fibrosis and in vitro on epithelial cell apoptosis and fibroblast migration and differentiation; however, none have advanced to or beyond clinical trials. Similar to LPA_1_, deficiency of *Lpar2* in mice conferred protection from bleomycin-induced lung inflammatory injury and PF in mice [293]. Reduced number of TUNEL^+^ apoptotic alveolar and bronchial epithelial cells was observed in the lung tissues of bleomycin-challenged *Lpar2* mice compared to the WT mice. Further, LPA-induced expression of TGF-β and differentiation of HLFs was reduced in lung fibroblasts deficient of *Lpar2*. LPA was found to signal through *Lpar2* to regulate the ERK1/2, SMAD3, AKT, and p38 MAPK pathways, but not SMAD2 and JNK pathways during PF [293] (Figure 6). The role of LPA_3–6_ in IPF and experimental lung fibrosis is yet to be established.

### 5.3. Lipid Peroxidation and Oxidized Phospholipids in Pulmonary Fibrosis

Membrane lipids are rich in polyunsaturated fatty acids, which are susceptible to oxidation and peroxidation by hydroxyl (^●^OH), hydroperoxyl (^●^OOH) radicals, and ROS to generate lipid peroxidation end products, mildly oxidized and truncated oxidized phospholipids [308]. Lipids can also be oxidized by LPs, COXs, and cytochrome P450 to yield lipid peroxidation products, and oxidized phospholipids. Cellular antioxidant defense mechanisms maintain the normal redox status of the cell by scavenging, neutralizing, and repairing the free radicals and oxidized lipids; however, in pathological conditions, these lipid oxidation products accumulate and affect cell function and viability. Elevated levels of lipid peroxidation products such as 4-hydroxynonenal, oxidized phospholipids, or their protein adducts have been identified in BAL fluids and lung tissues of patients with IPF, COPD, and ARDS [309,310,311]. Generation and role for lipid peroxidation products have been shown in bleomycin-induced PF [312] and antioxidants such as α-tocopherol protected the mice against bleomycin-induced lung injury [313] suggesting a role for lipid peroxidation in pulmonary toxicity. Resveratrol, a phytochemical and antioxidant, also alleviated bleomycin-induced lung injury in rats [314] and the thiol protectant, *N*-acetylcysteine, and the iron chelator, deferoxamine attenuated the bleomycin-mediated oxidative stress and lung injury [315] supporting the findings that bleomycin-induced oxidative stress, altered thiol redox status, induced lipid peroxidation, activated PLD, and caused cytotoxicity in a redox-dependent pathway in lung endothelial cells [238]. Glutathione peroxidase (GPx) is an important antioxidant enzyme that reduces phospholipid hydroperoxides and modulation of GPx expression or activity would tilt the redox balance towards oxidation. Reduced expression of GPx4 and increased 4-hydroxynonenal immunostaining was seen in IPF lungs, and genetic deletion of GPx4 (*Gpx4^+/−^*) mice showed enhanced bleomycin-induced lung fibrosis and TGF-β-mediated myofibroblast differentiation in vitro [316]. Thus, changes in the redox status due to tress could play a critical role in the pathophysiology of IPF.

There is compelling evidence for monocyte-derived macrophages and not residential lung macrophages to be involved in the development of PF in experimental murine models [317]. Accumulation of lipid-laden AMs, also known as foam cells, has been characterized in experimental models of fibrosis and histological lung specimens from IPF patients [318,319]. A significant increase in oxidized-PC (Ox-PC) was observed in AMs and BAL fluid after bleomycin injury and Ox-PC, but not native PC, enhanced the expression of *Tgfβ1*, *Ym1*, *cd36*, *cd63,* and *cd56* transcripts indicating polarization of macrophages to M2 phenotype. Instillation of Ox-PC directly into the mouse lung induced foam cell formation and accumulation and development of PF. The increased accumulation of lipids in macrophages occurs through efferocytosis whereby macrophages engulf apoptotic alveolar type II epithelial cells after injury via CD36 on the macrophages [319,320,321]. Targeted deletion of the lipid efflux transporter, ATP-binding cassette sub-family G member 1 (*Abcg1*) increased foam cell formation and exacerbated lung fibrosis in mice from bleomycin challenge [320]. Thus, a pneumocytes-macrophage-CD36-oxidized lipid signaling axis has been proposed to play a key role in the development of lung fibrosis in the bleomycin model [319,322]. Another potential mechanism of Ox-PC in development of PF is through PAI-1. Administration of 1-palmitoyl-2-(5-oxo-valeroyl)-*sn*-glycero-3-phosphocholine (POVPC), an Ox-PC, to *Pai-1^+/+^* increased hydroxyproline in the lung and reduced serum SP-D levels in contrast to *Pai-1^−/−^* mice treated with POVPC [323,324]. These results suggest that PAI-1 promotes fibrosis in response to oxidized phospholipid administration in mouse lungs.

## 6. Conclusions

IPF is a heterogeneous interstitial lung disease caused by abnormal host-defense, activation of immune and non-immune cells, and dysfunctional wound, male gender, cigarette-smoking, and mutations in *MUC5B* and *Sftpc* gene confer a predisposition to the development of IPF. In the last three decades, several target genes and proteins have been identified as key players in the pathophysiology healing and lung repair that result in lung fibrosis. With the advancement in mass spectrometry and lipidomics, mediators derived from fatty acids, glycerolipids, phospholipids, and sphingolipids have been identified in lung tissues and biological fluids of IPF patients and experimental models of fibrosis. These bioactive lipid mediators exhibit pro- or anti-fibrotic effects in vivo and *in vitro*. PLA_2_ catalyzed hydrolysis of a phospholipid such as PC enriched in arachidonic acid generates free AA that is subsequently converted by COX2/PG synthase to prostanoids such as PGD2, PGE_2,_ PGF2α, and by 5-LO to LTs. While PGD2 and PGF2α and LTs have pro-fibrotic properties, PGE2 exhibits anti-fibrotic effects in the experimental bleomycin model of PF and TGF-β-induced fibroblast activation. The SPHK1/S1P/S1PR, PLD2/PA, and ATX/LPA/LPAR signaling axes have been identified as significant contributors to fibrosis in IPF patients and bleomycin/radiation models of PF. Blocking SPHK1 with SKI-II attenuated lung inflammation and development of lung fibrosis 3 weeks post-bleomycin challenge indicating a long-term beneficial effect of this SPHK1 inhibitor. However, PF543 is yet to be evaluated for its efficacy in clinical trials. As S1P signals via S1PRs, and deletion of *S1PR2* or *S1PR3* ameliorated bleomycin-induced PF, use of specific receptor antagonists for S1PR2 or S1PR3 might be beneficial, which needs further investigation.

There has been tremendous advancement in developing ATX inhibitor(s) as a potential therapy in IPF (Table 1). The ATX inhibitor from Galapagos Pharmaceuticals, GLPG1690 exhibited a good PK/PD profile in experimental animal model of PF and has entered Phase III clinical trials after completing the Phase II studies in IPF patients. Blocking LPA_1_ with antagonists such as BMS-976278, AM152, SAR100842, and ONO-7300243 conferred protection against bleomycin-mediated lung inflammation and fibrosis in vivo and in vitro on epithelial cell apoptosis and fibroblast migration and differentiation; BMS-976278 has shown promise in clinical trials. Another promising target is LYCAT, a cardiolipin remodeling enzyme, which is upregulated in the lungs of IPF patients and bleomycin treated animals. LYCAT overexpression in mouse lung reduced bleomycin-induced PF and activation of LYCAT activity or enhances its expression in lung epithelial cells and fibroblasts might be a novel approach to reduce PF in pre-clinical models before advancing to human studies. Other potential targets for drug development for IPF include cysLTs and its receptors. Montelukast, a cysLT type I receptor antagonist, approved by the FDA for the management of asthma. Although Montelukast showed some beneficial effects against IPF, and its potential as a therapeutic agent for lung fibrosis is inconclusive and require more trials. Another class of mediators known as lipoxins, resolvins, protectins, and maresins, which are derived from polyunsaturated fatty acids may modulate the resolution of inflammation and fibrosis. Resolvin D1 is one lipid mediator that is derived from eicosapentaenoic acid and docosahexaenoic acid attenuates lung inflammation and interstitial fibrosis as well as inhibits TGF-β-induced EMT suggesting it as a potential therapeutic agent against IPF. Thus, understanding the mechanisms of generation, and signaling pathways of lipid mediators generated in PF provides an opportunity to develop therapies to attenuate the development of lung fibrosis and facilitate the process resolution of fibrotic injury.

## Figures and Tables

**Figure 1 ijms-21-04257-f001:**
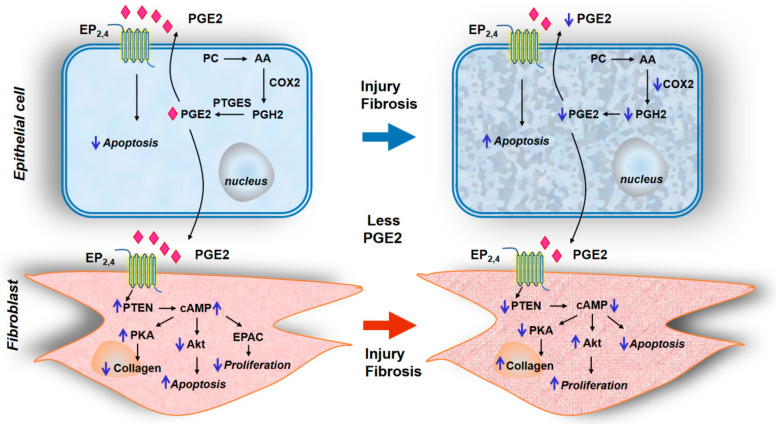
Prostaglandin E2 signaling via EP2/EP4 in epithelial cells and fibroblasts for the development of pulmonary fibrosis. Schema depicts PGE2 biosynthesis from arachidonic acid by COX2 and prostaglandin E synthases in alveolar epithelial cells. The autacoid function of PGE2 via its receptors EP2 and EP4 regulates homeostatic signaling between the alveolar epithelial cells (AECs) and pulmonary fibroblasts. Fibrotic signaling is specified by enhanced apoptosis and diminished secretion of PGE2 by AECs due to injury, which in turn promotes fibroblast proliferation, collagen deposition, and myofibroblast differentiation, distinct in pulmonary fibrosis. AA—Arachidonic acid, cAMP—cyclic adenosine monophosphate, COX2—Cyclooxygenase-2, EP_2,4_—Prostaglandin EP_2_,EP_4_ receptor, EPAC—Exchange protein directly activated by cAMP, PC—Phosphatidyl choline, PGE2—Prostaglandin E2, PGH2—Prostaglandin H2, PKA—Protein kinase A, PTGES—Prostaglandin E Synthase, PTEN—Phosphatase and tensin homolog.

**Figure 2 ijms-21-04257-f002:**
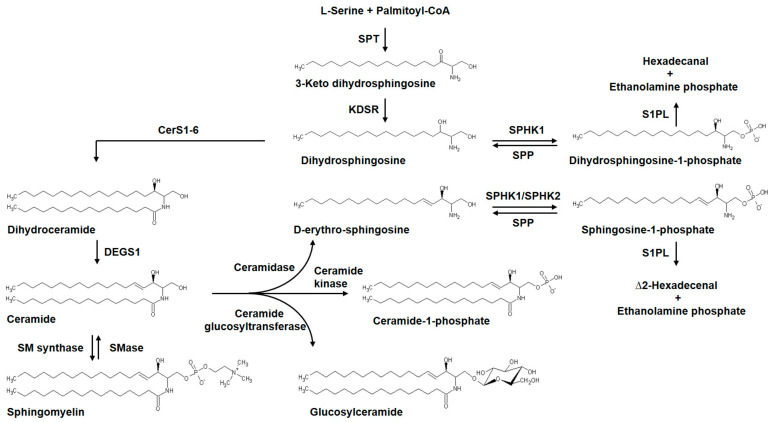
Sphingolipids implicated in pulmonary fibrosis, their structure, and metabolizing enzymes. An overview of de novo pathways of sphingolipid biosynthesis. Formation of the precursor 3-keto dihydrosphingosine from palmitoyl-CoA and L-serine catalyzed by SPT is the initial rate-limiting step, followed by generation of complex sphingolipids catalyzed by specific metabolic enzymes resulting in the generation of ceramides, sphingosine, sphingosine-1-phosphate (S1P), ceramide-1-phosphate and glucosyl ceramide. S1P or Dihydro S1P is hydrolyzed by S1PL to ∆2-hexadecenal or hexadecanal, respectively, and ethanolamine phosphate. S1P or dihydro S1P is also converted back to sphingosine by SPP. Dysregulation of the metabolic pathway intermediates is involved in the pathogenesis of pulmonary fibrosis. CerS1-6—Ceramide synthase 1-6, DEGS1—Delta 4-Desaturase Sphingolipid 1, KDSR—Keto dihydrosphingosine reductase, SPT—Serine palmitoyltransferase, SMase—Sphingomyelinase, SM synthase—Sphingomyelin synthase, SPHK1—Sphingosine kinase 1, SPHK2—Sphingosine kinase 2, S1P—Sphingosine-1-phosphate, SPP—Sphingosine-1-phosphate phosphatase, S1PL—Sphingosine-1-phosphate lyase.

**Figure 3 ijms-21-04257-f003:**
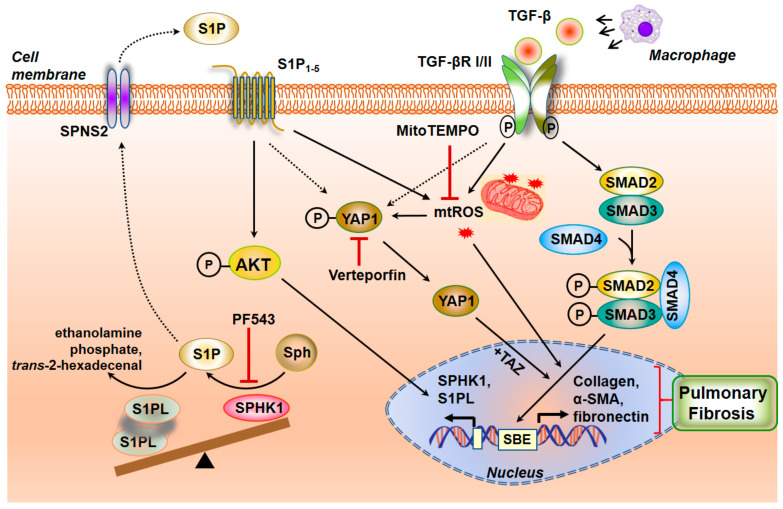
Crosstalk between S1P and TGF-β signaling cascade in pulmonary fibrosis. Profibrotic signaling by the inflammatory cytokine TGF-β is affected through the dimerization of TGF-β1 and TGF-βII receptors and its phosphorylation and activation of SMADs to induce a fibrogenic transcriptional program. Transport of S1P generated by SPHK1 activation from the cell to the extracellular milieu is enabled by SPNS2 transporter. S1P binding to S1P receptors stimulates mitochondrial reactive oxygen species (mtROS) generation and YAP1 translocation to the nucleus to influence myofibroblast transdifferentiation and matrix remodeling. AKT—Protein kinase B, αSMA—smooth muscle αActin, MitoTEMPO—(2-(2,2,6,6-Tetramethylpiperidin-1-oxyl-4-ylamino) -2-oxoethyl)triphenylphosphonium chloride, PF543—SPHK1 inhibitor, SBE—SMAD-Binding Element, SPNS2—spinster homolog 2, Sph—Sphingosine, SPHK1—Sphingosine kinase 1, S1P—Sphingosine -1-phosphate, S1PL—Sphingosine-1-phosphate lyase, S1P_1_–_5_—Sphingosine-1-phosphate receptors 1–5, SMAD—Mothers Against Decapentaplegic Homolog, TGF-β—Transforming growth factor beta, TGF-β RI/II—Transforming growth factor beta receptor I/II, TAZ—Tafazzin, YAP1—Yes-associated Protein 1.

**Figure 4 ijms-21-04257-f004:**
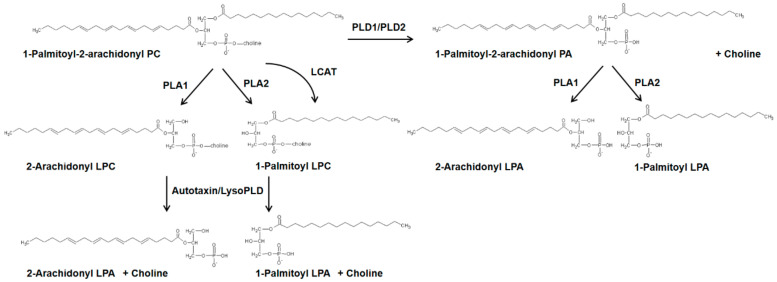
Generation of lysophosphatidic acid by two different pathways in mammalian cells. Schema showing the conversion of 1-Palmitoyl-2-arachidonyl phosphatidylcholine to LPA by the metabolizing enzymes, PLD, PLA_1_, PLA_2_, and autotaxin or lyso PLD. LCAT—lecithin-cholesterol acyltransferase, LPA—lysophosphatidic acid, LPC—lysophosphatidylcholine, lyso PLD—lysophospholipase D, PA—Phosphatidic acid, PC—Phosphatidylcholine, PLA1/2—Phospholipase A1/2, PLD1/2—Phospholipase D 1/2.

**Figure 5 ijms-21-04257-f005:**
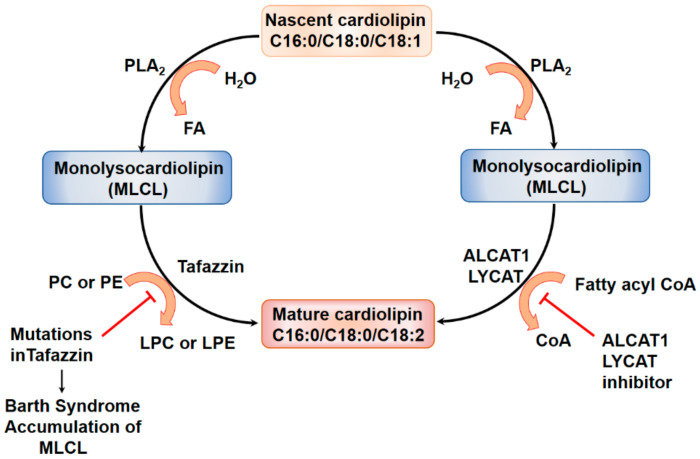
Cardiolipin remodeling by lysocardiolipin acyltransferase in pulmonary fibrosis. The mitochondrial phospholipid cardiolipin is converted by iPLA_2_ or cPLA_2_ to monolysocardiolipin (MLCL), a regulator of apoptosis, in the mitochondria. Dysregulation of LYCAT function, which remodels the fatty acid composition of cardiolipin, has been implicated in pulmonary fibrosis and Barth syndrome. ALCAT1—Acyl-CoA: lysocardiolipin acyltransferase, CoA—CoenzymeA, FA—Fatty acid, LYCAT—lysocardiolipin acyltransferase, LPC—lysophosphatidylcholine, LPE—lysophosphatidylethanolamine, PC—Phosphatidylcholine, PE—Phosphatidylethanolamine, iPLA_2_—calcium-independent Phospholipase A2, cPLA_2_—cytosolic phospholipase A_2_.

**Figure 6 ijms-21-04257-f006:**
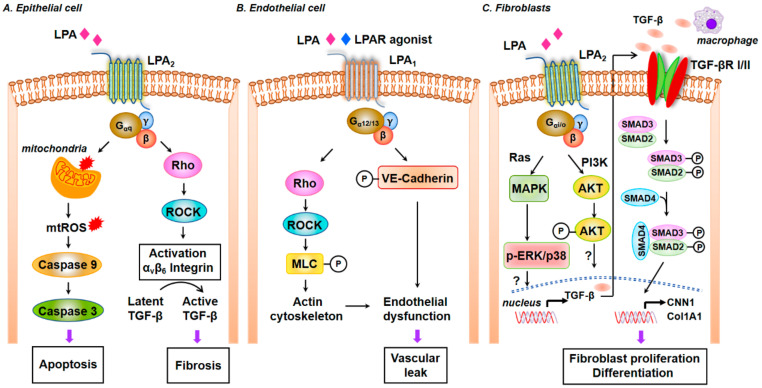
Role of lysophosphatidic signaling via lysophosphatidic acid receptors in pulmonary fibrosis. (**A**), Alveolar epithelial cell stimulation by lysophosphatidic acid (LPA) activates the G-protein coupled receptor LPA_2_ and generation of mitochondrial reactive species (mtROS) promotes caspase-3 mediated apoptotic pathway. Rho/ROCK activation by LPA directs the fibrogenic signaling via α_v_β_6_ and TGF-β. (**B**), In the lung vascular endothelial cells, the LPA/LPA_1_ activation leads to cytoskeletal changes and VE-cadherin-mediated junctional disruption. (**C**), LPA/LPA2 signaling in the fibroblasts stimulates the MAPK/ERK and PI3K/AKT pathways, resulting in enhanced TGF-β production and secretion. This results in activation of the TGF-β dependent profibrotic signaling mechanisms via TGF-βI/TGF-βII receptors and activation of SMAD proteins leading to myofibroblast differentiation and collagen production in PF. AKT—Protein kinase B, CNN1—Calponin1, Col1A1—Collagen Type I Alpha 1 chain, ERK—Extracellular signal-regulated kinase, G_α_q,12/13,i/o—GTP-binding protein complex alpha subunit, LPA—lysophosphatidic acid, LPA1/2—lysophosphatidic acid receptor type1/2, LPAR—lysophosphatidic acid receptor, mtROS—mitochondrial reactive oxygen species, MAPK—mitogen-activated protein kinase, MLC—myosin light chain, PI3K—Phosphatidylinositol-3-kinase, Ras—RAS family of GTPases, Rho—Rho family of small GTPase, ROCK—Rho-associated coiled-coil containing protein kinase, SMAD2/3/4—Mothers Against Decapentaplegic Homolog 2/3/4, TGF-β—transforming growth factor beta, TGF-β RI/II—transforming growth factor beta receptor I/II, VE—Cadherin vascular endothelial cadherin.

**Table 1 ijms-21-04257-t001:** Clinical studies to evaluate new drugs targeting lipid-mediated pathways in the treatment of pulmonary fibrosis.

Drug	Target Specificity	Identifier ^a^	Stage
BMS-986278	LPA_1_ receptor antagonist	NCT04308681	Phase II
BMS-986020	LPA_1_ receptor antagonist	NCT01766817	Phase II
18F-BMS-986327	LPA_1_ ligand for PET	NCT04069143	Phase I
Tipelukast/MN-001	Orphan drug (LT receptor, PDE inhibitor, 5-LO pathway)	NCT02503657	Phase II
GLPG1690	ATX inhibitor	NCT03733444	Phase III
BBT-877	ATX inhibitor	NCT03830125	Phase I
X-165	ATX inhibitor	Preclinical, IND	-
GSK2126458	PI3K/mTOR inhibitor	NCT01725139	Phase I
Sirolimus	mTOR inhibitor	NCT01462006	Phase I
HEC68498	PI3K/mTOR inhibitor	NCT03502902	Phase I

^a^ Unique identifier from https://clinicaltrials.gov/; ATX—Autotaxin; IND—Investigational new drug; LPA_1_—Lysophosphatidic acid receptor type1; 5-LO—5-lipoxygenase; LT—Leukotriene; mTOR—mammalian target of rapamycin; PDE—Phosphodiesterase; PET—Positron emission tomography; PI3K—Phosphoinositide 3-kinase.

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
