# Peer review of "Lipid Mediators Regulate Pulmonary Fibrosis: Potential Mechanisms and Signaling Pathways"

_ijms, 2020, doi:10.3390/ijms21124257_

Round 1

Reviewer 1 Report

In this review manuscript, the authors extensively describe the state of the art literature on bioactive lipids regulating mechanisms leading to pulmonary fibrosis in preclinical cell culture and animal models and in patients suffering from IPF.

Major points:

1-The review is partly exhaustive and it is highly recommended to shorted the text. Partly, studies are described in too much detail and all results just repeated from the original publication.

2-It would be nice to have a table with an overview on all clinical studies (phase 1-3), including NCT numbers, which target lipid pathways. This would make it more clear, which lipid pathway is most promising to yield new therapeutics. As obviously the ATX/LPA seems to be the most advanced, it is recommended to start with this lipid class first.

3-On lines 478-479, When refering to S1P acting on TRAF2, it would be fair enough to mention that these data are not reproduced by others. See Etemadi et al. 2015.

4-On lines 606-608, the authors state that myriocin downregulated S1PL protein expression, but had no effect on activity level. That is very surprising and confusing as it suggests that the same activity is see when less S1PL is present. Please recheck.

5-On lines 652-653, the authors state :   « … that FTY720-phosphonate, ponesimod and SEW2871 decrease levels of CTGF….. ». However, this is not what was reported by Sobel et al (ref 176). They show that S1P and FTY720-P (phosphate) robustly induce ECM synthesis and a set of fibrotic genes. Ponesimod was less active and SEW2871 was inactive.

6-Notably, one of the reported adverse effects of fingolimod was indeed that pulmonary function (FEV) was reduced. This observation would be interesting to include.

7-On lines 822-824 : Since a review should not include unpublished own data, this section should be eliminated.

Author Response

Response to Reviewer’s Comments

Reviewer 1:

General Comment: In this review manuscript, the authors extensively describe the state of the art literature on bioactive lipids regulating mechanisms leading to pulmonary fibrosis in preclinical cell culture and animal models and in patients suffering from IPF.

Response: We thank the reviewer and appreciate the positive comments.

Major points:

Comment1: The review is partly exhaustive, and it is highly recommended to shorten the text. Partly, studies are described in too much detail and all results just repeated from the original publication.

Response: This is a valid comment and accordingly the text content has been reduced wherever possible. Due to a dearth in reviews covering the role of Lipid mediators in IPF and animal models of pulmonary fibrosis, we have taken an approach to provide more details here. Further, we have covered diverse lipid-mediated signaling pathways in this review article.

Comment 2: It would be nice to have a table with an overview on all clinical studies (phase 1-3), including NCT numbers, which target lipid pathways. This would make it more clear, which lipid pathway is most promising to yield new therapeutics. As obviously the ATX/LPA seems to be the most advanced, it is recommended to start with this lipid class first.

Response: This is an excellent suggestion. We have included a Table in the Conclusion section that lists the clinical study, targets and NCT numbers.

Comment 3: On lines 478-479, When referring to S1P acting on TRAF2, it would be fair enough to mention that these data are not reproduced by others. See Etemadi et al. 2015.

Response: This is a fair comment and as suggested we have added the following sentence “However, in keratinocytes, deletion of Traf2, but not Sphk1, disrupted TNF-α-mediated NF-kB and MAPK signaling causing skin inflammation [Etemadi N et al., eLife 2015; 4: e10592; doi: 10,7554/eLife. 10592]”.

Comment 4: On lines 606-608, the authors state that myriocin downregulated S1PL protein expression, but had no effect on activity level. That is very surprising and confusing as it suggests that the same activity is see when less S1PL is present. Please recheck.

Response:  The statement that “myriocin downregulated S1P lyase protein expression, but had no effect on its activity” is correct as stated in the published article. However, we have modified the sentence as “Myriocin treatment attenuated radiation-induced increase in expression of SPHK1, SPT and SGPL1, but not SPHK2; however, the activity of S1P lyase was not modulated compared to control animals”.

Comment 5: On lines 652-653, the authors state:   « … that FTY720-phosphonate, ponesimod and SEW2871 decrease levels of CTGF….. ». However, this is not what was reported by Sobel et al (ref 176). They show that S1P and FTY720-P (phosphate) robustly induce ECM synthesis and a set of fibrotic genes. Ponesimod was less active and SEW2871 was inactive.

Response: Thanks for catching this inadvertent statement. This sentence has been modified to “S1PR1 and S1PR3 agonists such as FTY720-phosphate, and Ponesimod, but not SEW2871, caused a robust stimulation of extracellular matrix (ECM) synthesis and expression of pro-fibrotic genes including CTGF [Ref]”.

Comment 6: Notably, one of the reported adverse effects of fingolimod was indeed that pulmonary function (FEV) was reduced. This observation would be interesting to include.

Response: As suggested by the reviewer, we have included this additional statement “Fingolimod (FTY720) at 1.25 mg and 5.0 mg daily dose given to patients with relapsing multiple sclerosis showed a reduction in pulmonary function (Kapposz et al., NEJM 2006; 355: 1124-1140) as observed in IPF patients”.    

Comment 7: On lines 822-824: Since a review should not include unpublished own data, this section should be eliminated.

Response: As suggested, we have deleted the sentence(s) on the unpublished data.

Reviewer 2 Report

Although the authors have written a long manuscript, many aspects were not addressed.
There are no basic informations about pathogenesis with aproppriate citations.
There are no links to other aspects of disease development.
No information about the role of mesenchymal transdifferentiation (EMT, EndMT) in the pathogenesis.
There is no information about the role of other members of TGF-b family:TGF-b2 TGF-b3 in the disease deveopment.
The authors described only selected Smad-independent paths. All known pathways should at least mentioned.

Author Response

Response to Reviewer’s Comments

Reviewer 2:

General Comment: Although the authors have written a long manuscript, many aspects were not addressed.

Response: IPF and pulmonary fibrosis is a widely studied topic with several targets, multiple canonical and non-canonical pathways involving epithelial cells, endothelial cells, fibroblasts and immune cells. Many excellent reviews have highlighted the mechanisms involved in the progression of this devastating disease with a goal towards drug development. Reviewer #1 has suggested shortening the article. For the purpose of this review, we focused more on our current understanding of lipid-mediated signaling pathways in the pathogenesis of IPF/PF hoping to do justice on a specific topic.

Comment 1: There is no basic information about pathogenesis with appropriate citations.

Response: As suggested, we have added additional sentences on the pathogenesis of pulmonary fibrosis under “Introduction” in the revised manuscript.

Comment 2: There are no links to other aspects of disease development.

Response: This is an important suggestion; however, this review addressed the role of lipid mediators in the pathogenesis of pulmonary fibrosis and we have included available information from the literature at appropriate sections. 

Comment 3: No information about the role of mesenchymal transdifferentiation (EMT, EndMT) in the pathogenesis.

Response: As per the suggestion, a couple of sentences on the role of EMT and EndMT in the pathogenesis of IPF/pulmonary fibrosis are added to the text under “Introduction” and the section on TGFβ/S1P signaling in PF.

Comment 4: There is no information about the role of other members of TGF-b family:TGF-b2 TGF-b3 in the disease development.
Response: Although three isoforms of TGF-β, namely TGF-β, -2, and -3 been reported, TGF-β1 is the most pre-dominant isoform expressed in the normal lung and its expression is enhanced in IPF and animal models of pulmonary fibrosis. Most of the studies related to TGF-β is due to the action of TGF-β1 isoform and practically no information regarding the role of TGF-β2 and TGF-β3 signaling in IPF and experimental pulmonary fibrosis is described in the literature. This information has been added to the Introduction paragraph in the revised manuscript.

Comment 5: The authors described only selected Smad-independent paths. All known pathways should at least be mentioned.

Response: We have addressed this concern in the revised manuscript under “Introduction” -para 3.

Round 2

Reviewer 2 Report

Despite the long time of corrections, the manuscript is still poorly written. The introduced corrections regarding important processes in the discussed issues are very short and the authors' explanations are quite vague. To conclude, the manuscript has a moderate quality and in my opinion, should be rewritten.

Author Response

Unlike a comprehensive review on interstitial lung disease and IPF, it is our intention in this review to cover most aspects of studies involving lipid and lipid-mediated signaling pathways with regards to their role in the progression of pulmonary fibrosis. Sections 2-5 describe in an exhaustive manner, studies involving specific class of lipids/lipid-intermediates/lipid-mediators which have been/are targeted using preclinical models with relevance to the human disease. We recognize that one review like this cannot do justice to all aspect of pathogenesis of PF. We have included a statement in the third para under “Introduction” acknowledging inability to touch on all of them.